# A Lightweight Three-Factor Authentication Scheme for WHSN Architecture

**DOI:** 10.3390/s20236860

**Published:** 2020-11-30

**Authors:** Abdullah M. Almuhaideb, Kawther S. Alqudaihi

**Affiliations:** Department of Computer Science, College of Computer Science and Information Technology, Imam, Abdulrahman Bin Faisal University, P.O. Box 1982, Dammam 31441, Saudi Arabia; amalmuhaideb@iau.edu.sa

**Keywords:** WHSN, BAN logic, tamarin prover, authentication protocol, stolen smart card, anonymity

## Abstract

Wireless Healthcare Sensor Network (WHSN) is a benchmarking technology deployed to levitate the quality of lives for the patients and doctors. WHSN systems must fit IEEE 802.15.6 standard for specific application criteria, unlike some standard criteria that are difficult to meet. Therefore, many security models were suggested to enhance the security of the WHSN and promote system performance. Yu and Park proposed a three-factor authentication scheme based on the smart card, biometric, and password, and their scheme can be easily employed in three-tier WHSN architecture. Furthermore, they claimed that their scheme can withstand guessing attack and provide anonymity, although, after cryptanalysis, we found that their scheme lacks both. Accordingly, we suggested a three-factor authentication scheme with better system confusion due to multiplex parametric features, hash function, and higher key size to increase the security and achieve anonymity for the connected nodes. Moreover, the scheme included initialization, authentication, re-authentication, secure node addition, user revocation, and secure data transmission via blockchain technology. The formal analysis of the scheme was conducted by BAN logic (Burrows Abadi Nadeem) and the simulation was carried out by Tamarin prover to validate that the proposed scheme is resistant to replay, session hijacking, and guessing attacks, plus it provides anonymity, perfect forward secrecy, and authentication along with the key agreement.

## 1. Introduction

Wireless Sensor Network (WSN) is widely spread through various firms such as shrewd homes, shrewd manufactory, shrewd businesses, and smart health systems such as in WHSN [1,2,3,4,5,6,7]. This technology aims to reduce the patient’s need to go to the hospital for checkups and allow the doctors to monitor the patients’ health status from a remotely far location at any time. In the latest years, the adaptability of WHSN consists of small sizes, lower power, cheap sensors, and enables the communication among them to occur in a short-range [8]. Those sensors can be micro-controller, transceiver, memory, and battery. WHSN architecture supports sensors cooperation with each other’s to build the connected sensor network architecture and inspect the user’s health [9], as depicted in Figure 1.

The data collected by the sensor are saved for long time to increase its quality and to make better processing and analysis for better treatment choices [10]. Also, WHSN architecture consists of weak sensors that infringe the privacy of the patient data. Many authentication schemes were proposed to solve this issue along with many others such as anonymity, eavesdropping, DoS (Denial of Service Attack), and nodes impersonation attack [11]. After thorough analysis for the proposed schemes, we found that each has its strengths and weaknesses.

Recently, Yu and Park [12] proposed three-factor authentication scheme (SLUA-WSN) for WSN network smart homes to enable the user of authenticating themselves in a secure manner. They claimed that their scheme is protected against impersonation, stolen integrated circuit card, and guessing attacks, and provides user-anonymity with un-traceability. However, we identified a lack of smart card data protection that leads to node impersonation and guessing in cases where stolen smart card attack occurred. Also, issues in anonymity and un-traceability arise, when all the previously mentioned acts are committed by the intruder i. Their scheme can be improved regarding computation and communication costs on both the foreign network side and gateway side too. Therefore, we propose a robust authentication scheme based on three-factor for WHSN higher performance and capacity efficiency besides advanced security to overcome the weaknesses in [12] scheme.

### 1.1. Contribution and Motivation

In continuation to the development of the WHSN authentication scheme that is proposed in our previous research [13]. We considered that the sensor node data is secure, and we proposed a secure authentication scheme between the foreign network node and the hub node. The main contributions of this article are as follows:Performing cryptanalysis of Yu and Park [12] scheme and show its vulnerability regarding anonymity protection, un-traceability protection, impersonation, guessing, and stolen smart card attacks.Proposing a lightweight three-factor authentication and re-authentication schemes consist of the biometric, smart card, and password with better key management, and less operations to increase the scheme efficiency. Also, introducing additional mechanisms such as secure node addition, secure user revocation, and data transmission via blockchain.Validate the scheme BAN logic, and Tamarin simulation tool to prove its authentication, key agreement, and security. The results validated the scheme security versus replay, and session hijacking attacks, plus it achieved perfect forward secrecy along with authentication and key agreement.Calculate the efficiency of the new scheme with computation in line with communication costs and storage. It showed an advantage of our scheme over [12] structure regarding computation cost, communication cost, and storage capacity.

The motivations of our work are described below:
The noticeable drawbacks in most of WHSN structures, and their weaknesses towards most well-known attacks such as impersonation, session hijacking, and stolen smartcard attacks.Designing authentication scheme needs to achieve system scalability along with security.WHSN authentication schemes must provide appropriate complexity algorithm in conjunction the system capabilities along with capacity.

Accordingly, we proposed a lightweight authentication scheme to enhance the security and solve the performance deficiencies in [12]. The newly proposed scheme will provide more security with less hash functions and high parameters confusion. It is secure against offline/online shared secret guessing, brute force, replay, impersonation, eavesdropping, collision, and jamming attacks. Also, it provides numerous security features such as anonymity, integrity, un-traceability, key agreement, and mutual authentication. Likewise, the scheme is appropriate for WHSN constraint system due to its efficiency in comparison to other authentication schemes.

### 1.2. Organization

The remaining of this article is structured as follows. We present the state-of-art for WSN architecture in Section 2 and explain the preliminaries in Section 3. Section 4 analyzes Yu and Park’s structure, and Section 5 illustrates a protected and efficient authentication schemes for WHSN architecture to improve the downfalls of Yu and Park’s scheme. Section 6 assesses the security evaluation of the new scheme by executing informal and formal analysis containing BAN logic along with Tamarin simulation. Section 7 shows the outcomes of the efficiency analysis of the new scheme in comparison with the associated schemes. Finally, the conclusion is discussed in Section 8.

## 2. Related Works

In recent years, numerous access control and authentication systems were suggested to secure the data in WHSN technology. Some schemes are non-cryptographic based schemes that rely on the physiological signal, channel-based schemes that rely on special software or sensor, and cryptographic based schemes which are more popular [14].

Chang et al. and Park et al. [15,16] had offered an authentication structure between the user node and the gateway node and utilized a honeyword checker for the password security. Also, their scheme used random number generator from the Elliptic curve along with a hash function right before sending the authentication request. Consequently, C. Wang et al. [17] had cryptanalyzed both schemes and exposed their lack of anonymity along with their vulnerability to known session-specific temporary information (KSSTI), and privileged-user attacks. Therefore, ref. [17] suggested an improved anonymity three-factor authentication scheme utilizing an Elliptic curve cryptosystem (ECC). The structure relied on the biometric fuzzy extractor method to enhance scheme security against password guessing and identity spoofing. Unfortunately, their scheme suffered from issues in anonymity as well as backward secrecy attack when the user loses his/her smartcard, and due to some parameters lack protection.

Similarly, Challa et al. [18] recommended an authentication system with three factors in wireless body area network (WBAN) architecture based on the pubic key and Elliptic curve structure to create a secure system. They declared that their system is strong versus several types of attacks such as insider attack, password cracking, stolen smart-card, denial-of-service, known session key, masquerading, session hijacking, and replay attacks. However, their scheme lacked anonymity of the user and sensor identities. Also, the weak protection to the public key by the user phone and temporary identity made the scheme weak toward anonymity and guessing attack due to the exposure of random parameters in the open channel.

Mo and Chen [19] had analyzed the security flaws in the proposed three-factor scheme in WSN by Lu et al. [20] and found that their structure is suspectable to offline password cracking, known session-specific temporary information attacks, and lack of session key backward secrecy. Therefore, ref. [19] had offered a three-factor authentication structure utilized user biometric, smart card and key where they used hash function and Elliptic curve (ECC) to protect the passwords and security parameters. But the issue is the user anonymity might be compromised because the user identity is only protected by random number and biometric which both might be easily guessed and spoofed by the intruder i.

To deal with the sensitivity of data issues, Garg et al. [21] proposed a system using the Elliptic curve, signatures, and blockchain for WHSN to protect the transmitted data in an insecure channel and provide anonymity. Their scheme included the identity of the trusted authority as an additional secure parameter to authenticate between the communicated nodes. Although that their scheme deployed a great combination of cryptographic and emerging technologies to protect the data, it might face DoS attack and communication delay between the nodes, because of the heavy computation along with high storage cost. Ali et al. [22] had cryptanalyzed Liu and Chung [23] scheme and found out that it is unguarded to lost smart-card, offline key cracking, insider, and masquerading attacks. Moreover, ref. [22] had analyzed [18]’s approach, and found that it has correctness issues, broadcasting problems, lack of authentication between the trusted authority and sensor nodes, replay attack, Denial of Service (DoS), and forgery attacks.

Therefore, Ali et al. [22] had suggested a secure and lightweight three-factor authentication process for WHSN which employed both ECC, and bilinear pairing to resolve the issues in [18,23] schemes. Although their scheme is guarded against impersonation, privileged-insider, offline password cracking, stolen smart-card, and replay attacks, but it still has high computation cost and delays in communication due to extensive cryptographic operations.

One of the significant issues that faces the IoT authentication structures is jamming attack, when the intruder i sends jamming signal during the update of authentication values, and parameters [24]. In this context, two authentication schemes proposed by Shen et al. and Tewari et al. [25,26] that employed simple operations such as hash, XOR, and random number generators. Their schemes focused on time duration of the session, mutual random number generation, and keeping the latest identities of the communicating entities to increase the protection against jamming attacks.

Recently, Yu and Park [12] proposed SLUA-WSN which is a lightweight three-factor authentication scheme with secure user authentication system. Their scheme has the best in efficiency of all the previous schemes in the state-of-art, and the best robustness against sensor node capture, replay attack, insider attack, and impersonation attack, also it guarantees un-traceability and mutual authentication. Thus, SLUA-WSN is appropriate for applied WHSN environments because it is the strongest and efficient than related schemes. Their scheme suffers from stolen smart cards and shared secret key guessing because of the number of stored parameters in the smartcard. Secondly, there is no mechanism to check the validity of the generated random number at the first communication session between FN and GW. There must be a validation method to check whether the user generated the accepted random number that is generated before or not in case of mobile lost/smart card lost attacks.

## 3. Preliminary

This section deliberates the preliminaries used in both of our proposed protocols.

### 3.1. Fuzzy Extractor

In this section, we discuss fuzzy extractor function which is a cryptographic authentication mechanism that employs biometric and consists of two operations:Gen: After the biometric input Bio is imprinted by users, Gen produces a consistent random string σ {0, 1}, a random auxiliary string σ {0, 1}, and a probabilistic function.Rep: It reproduces σ with value σ when a disruptive biometric BIO new is inscribed, where σ is a public replication value connected with Bio.

### 3.2. Intruder Model

To analyze our model security, we discussed a very well-known Dolev–Yao (DY) threat model [27]. In the DY design, the intruder i capabilities are as presented below.
Referring to the DY model [27] an intruder i can inject, delete, intercept, and eavesdrop the data exchanged over wireless networks.Using the DY model [27], the data transmitted over wireless networks can be implanted, modified, recorded, and snooped by an intruder i.An intruder i can capture legal users’ smart cards and can use power-analysis to retrieve confidential keys stored in memory [28].An intruder i can undertake numerous attacks after extracting the smart card’s secret credentials, such masquerade, offline key guessing, trusted insider, and forward secrecy attacks [29,30].

## 4. Review on Yu and Park Scheme

In this section, we reviewed [12] to discover its weaknesses and points of enhancements, also we conducted cryptanalysis of the scheme, and we found that it lacks anonymity and the protection against secret shared key guessing. We discuss the scheme symbols in Table 1 as well as the registration and authentication phases.

### 4.1. Registration Phase of Yu and Park Scheme

In the registration phase of [12], the user and GWN communicate with one another to produce username, password, biometric, and smartcard values:The user $U_{i}$, inputs his/her password, username, and biometric, extract the biometric features using reproduction function and send those value over a secure channel to GWN.GWN produces random value $r_{g}$, calculates identification values $MID_{i}$, $X_{i}$, $Q_{i}$, and $W_{i}$ to store {*Qi*, *Wi*, *MIDi*} in the SC, and save $r_{g}$ in secure database. The number of saved parameters in the smartcard causes a weakness, that the attacker can seize to exploit the system parameters, by performing a smartcard impersonation attack along with database hijacking to retrieve the random value and user biometric.

### 4.2. Authentication Phase of Yu and Park Scheme

In the authentication phase of [12], the user and GWN authenticate each other along with the sensor to agree on the next session key as follows:Client $U_{i}$ inputs his/her username, password, and biometric in the smartphone, and checks the user identity before generating a current random value $R_{u}$ along with a timestamp.The mobile masks the following parameters such as original user identity $ID_{i}$, the hidden user identity $SID_{i}$, user masked identity in the smartcard $MID_{i}$, and the masked $X_{i}$.The GWN node receives the parameters, checks the timestamp to avoid replay attack, and retrieves the random values from the masked $X_{i}$ without checking if the random value had been used before. This step might rise a vulnerability in the scheme in case of a user node impersonation attack.The GWN shares the hidden values with the sensor $S_{j}$ for further authentication to the user node, and to support the next session key generation.$S_{j}$ authenticates both GWN and $U_{i}$, and generates new random nonce to produce the next pre-shared key.Both GWN and $U_{i}$ receive the new parameters to recover the generated values and save the new session key.

### 4.3. Cryptanalysis of Yu and Park Scheme

From the above, we deliberated the flaws for [12] structure in both registration and authentication phases which are: The number of authentication parameters in the smartcard along with the absence of random number checking in the GWN. Those two weaknesses allow the intruder to impersonate the user in the lost smartcard attack and weaken the anonymity.

#### Stolen Smart Card Attack

In this malicious act, an intruder i may attempt to masquerade the legal client and discover all the security parameters of the user by stealing the user smartcard and performing a guessing attack. According to the intruder model, we assumed that intruder i had extracted all the secret credentials that are adequately enough to impersonate the user from the smart card as follows $\left\{Q_{i},\;W_{i},\;MID_{i}\right\}$ and had obtained the random number $\left\{r_{g},\;K_{GWN}\right\}$, and spoofed the BIO of the user by performing database hijacking attack on the GWN node and smartphone node. Then, the attacker can perform the following:Intruder i computes the $ID_{i}=h(MID_{i}∥\;h\left(K_{GWN}∥r_{g})\right)$, and discovers the identity of the real user that brings lack of anonymity issue.Calculate $X_{i}=h\left(MID_{i}∥\;r_{g}∥K_{GWN}\right)$, $MPW_{i}=h\left(MID_{i}∥\;X_{i}\right)\;⊕Q_{i}$.Start authentication operation use the spoofed BIO and evade the threshold value of the biometric checking.$U_{i}\;$ computes the security parameters $X_{i},\;W_{i}^{*},\;MPW_{i}$, then checks the $W_{i}^{*}=?\;W_{i}$ provided by the attacker. If they are the same then the attacker can generate any random value instead of $R_{u}$ which is $R_{fake}$ and current time $T_{1}$ to deduce the following: $CID_{fake}=\left(ID_{i}∥SID_{j}\right)\;⊕h(MID_{i},\;R_{fake})$, $M_{UG}=h\left(ID_{i},\;R_{fake},X_{i},\;T_{1}\right)$, then send the following parameters to GWN $\left\{M_{1},\;MID_{i},\;CID_{fake},\;M_{UG},\;T_{1}\right\}.$GWN first checks the timestamp, if it is valid, then it will check other security parameters without validating the random number $R_{fake}$ whether similar to the random $R_{u}$ in the database or whether it is used before or not. Moreover, if the GWN does not have a mechanism to check the validity of the random number generated by the $U_{i}$ node then the Intruder i can generate fake random values and bypass the system security.Then GWN chooses any random number $R_{g}$ the same random number known by the intruder i, and the current time $T_{2}$ to calculate $M_{2},\;M_{GS}$ by performing the following equations $M_{2}=\left(R_{fake}∥R_{g}\right)⊕h\left(SID_{i}∥X_{j}∥T_{2}\right)$ and $M_{GS}=h\left(MID_{i}∥SID_{j}∥\;R_{fake}∥R_{g}∥T_{2}\right)$. Then, GWN sends the parameters $\left\{M_{2},\;MID_{i},\;M_{GS},\;T_{2}\right\}$ to $S_{j}$.Intruder i can intercept the parameters $\left\{M_{2},\;MID_{i},\;M_{GS},\;T_{2}\right\}$ between the channels to get the sensor node identity by applying this formula $SID_{j}=h\left(MID_{i}∥M_{GS}∥\;R_{fake}∥R_{g}∥T_{2}\right)$.

After repeating these steps from 1 to 7 by the intruder i, the intruder can discover all the security parameters alongside predict, intercept, and impersonate all the next upcoming parameters between the channels. This cryptanalysis showed that a smart card attack with little effort from the attacker can jeopardize the nodes anonymity and evade the system security.

In the next section, we propose the enhanced protocol to improve the security of [12] scheme that covers different phases in the authentication process.

## 5. Proposed Protocol

We explain our suggested authentication scheme assuming that our sensor node is trusted node and we want to secure the communication between the hub node and foreign network node. As the foreign network node works as a data collector for the sensor node and ensures the security of the transmitted parameters to the hub node. The scheme ensures strong authentication between FN-HN due to its three-factor authentication nature, and complex parametric system. In the below explanation, we showed the system five phases such as FN pre-deployment and registration, FN-HN authentication, re-authentication, safe node addition, user revocation, and secure data transmission via blockchain. Refer to Table 2 below for the notations.

### 5.1. Initialization Stage

In this stage, the parameter generation and registration of this protocol engaged the SM to choose secret identities, parameters and keys and allow all the entities to share securely the generated arguments over an offline and secure channel to *SN*, *FN*, and *HN*:The SM stores the $ID_{SN}$, $ID_{FN}$, and $ID_{HN}\;$ in the SM memory.SM chooses secret key $K_{MS}$, and l∗  as a secret parameter to be added to the node’s keys for the confusion.SM computes the secret key between the parameters
(1)$$SK=h\left(ID_{SN}\;∥h\left(l*\;∥K_{MS}\right)\;∥\;ID_{FN}∥\;ID_{HN}\right)$$SM calculates a new shadow identity for all the communicated nodes to ensure their anonymity during the communication and transmits those identities over a secure channel.
(2)$$ID_{SN}^{+}=ID_{SN}\;∥\;l*\;∥ID_{HN}$$
(3)$$ID_{FN}^{+}=ID_{FN}\;∥l*\;∥ID_{HN}$$
(4)$$ID_{HN}^{+}=ID_{SN}\;∥\;l*\;∥ID_{HN}\;∥\;ID_{FN}$$*HN* communicates with *SN* to generate secret parameters in a secure channel to authenticate each other during the communication.(5)$$E=h\left(ID_{SN}\;∥h\left(l*\;∥ID_{HN}^{+}\right)\right)$$*SN* saves the newly generated secret parameter in a secure location.*HN* communicates with *FN* to generate secret parameters in a secure channel to authenticate each other during the communication.(6)$$F_{i1}=h\left(ID_{FN}\;∥h\left(l*\;∥ID_{HN}^{+}\right)\right)$$*FN* keeps the new produced secret parameter in a secure location to enable the user from registering securely in the WHSN authentication system.

### 5.2. Registration Phase

In this phase, the client uses his/her smartphone to enter the password, identity, and biometric to allow the *HN* from generating an authentication smart card securely:The user inputs his/her $ID_{u}$ and $PW_{U}$ and imprints the biometric BIO to compute the user identity and password for SC registration:(7)$$Gen\left(BIO\right)=\langle \delta_{i}|\tau_{i}\rangle$$
(8)$$SPW=h\left(BIO\;∥ID_{FN}^{+}∥\;\delta_{i}\right),$$
where $\delta_{i}$: is the user biometric feature and $\tau_{i}$: Is the threshold. Then, *FN* sends these values $\left\{ID_{U},\;SPW\right\}$ to *HN* in a secure channel.*HN* receives the parameters, generates random value *rand*, and computes the following:(9)$$SID_{U}=h\left(ID_{U}\;∥ID_{FN}^{+}∥\;ID_{HN}^{+}∥\;rand\right)$$
(10)$$F_{\left(i+1\right)}=h\left(SID_{U}\;∥ID_{FN}^{+}∥\;rand\right)$$
(11)$$G=h\left(SID_{U}\;∥SPW∥\;l*\;∥rand\right)$$*HN* hides the value of *rand* with this equation:(12)$$H=h\left(rand\;∥ID_{FN}^{+}∥\;ID_{HN}^{+}\;∥ID_{SN}^{+}\right)$$Store the parameters $\left\{F_{\left(i+1\right)},\;G,\;H\right\}$ in the SC and send it to the user.*FN* receives the parameters and retrieves the random number from Formula (12) to store it securely in the memory and deletes H from the smart card to avoid stolen smart card attacks. The set of new parameters will be $\{F_{\left(i+1\right)},\;G$}, as depicted in Figure 2.

### 5.3. P-I: Authentication Phase

In this section, we assumed that the *SN* is a trusted node and *FN* authenticates itself and *SN* to the *HN*. Furthermore, it encompasses four phases of communications including *FN*, *HN*, and *SN* depicted in the Figure 3 below and denoted as follows:


**Step 1:**
$FN\;\to \;HN\;\left(M1=\left\{F_{\left(i+1\right)},\;A,B,\;t_{i1}\right\}\right)$
The user inserts the smart card, enters his/her $ID_{u}$, $PW_{U}$, imprints the biometric BIO, and calculates $Gen\left(BIO\right)=\langle \delta_{i}|\tau_{i}\rangle$ from Formula (7) and $SPW=h\left(BIO\;∥ID_{FN}^{+}∥\;\delta_{i}\right)$ from Formula (8). Also, computes $G^{*}=h\left(SID_{U}\;∥SPW∥\;l*\;∥rand\right)$ from Formula (11).*FN* checks whether $G^{*}=?G$, then continue, else abort the session. *FN* generates a new timestamp $t_{i1}$ random value $rand^{*}$ and calculates the following:(13)$$A=\;rand^{*}\;⊕\;F_{i1}⊕ID_{SN}^{+}$$
(14)$$B=\;Rand^{*}\;⊕SK$$


The previous step is very important to prevent jamming attack in any communication session. It allows both *FN* and *HN* to be part of generating random numbers that supports the session key formation.

*FN* sends the parameters to the *HN* in the open channel $\left\{F_{\left(i+1\right)},\;A,\;B,\;t_{i1}\right\}$.


**Step 2:**
$HN\;\to \;FN\;\left(M2=\;\left\{J,\;K,\;t_{j2}\right\}\right)$


*HN* performs the following:Verify the time $\Delta t=t_{j2}-t_{i1}$ to prevent a replay attack.If $\Delta t=t_{j2}-t_{i1}>0$ then continue, else terminate the process.Calculate $SID_{U}^{*}=h\left(F_{\left(i+1\right)}\;∥ID_{FN}^{+}∥\;rand\right)$ from Formula (10).Check if $SID_{u}^{*}=?\;SID_{U}$ to validate the user identity and remain, else terminate the process.*HN* should keep track of each used random number in the scheme to avoid replay attack or impersonation attacks.Calculate $rand^{*}\;=\;A\;⊕\;F_{i1}⊕ID_{SN}^{+}$, from Formula (13) and check if the $rand^{*}\;$ has been used before, if it is not continue to extract $ID_{SN}^{+}\;=\;A\;⊕\;F_{i1}⊕\;rand^{*}$, and check if the $ID_{SN}^{+}\;=?\;ID_{SN}^{+}$ to authenticate the sensor node.Calculate $F_{i1}=h\left(ID_{FN}\;∥h\left(l*\;∥ID_{HN}^{+}\right)\right)$ from Formula (6) to authenticate the *FN*.

$SK=\;rand^{*}\;⊕B$ from Formula (14) to validate the shared secret key.

After authenticating the *SN* and *FN*, *HN* generates new random nonce $rand^{+}$, new timestamp $t_{j2}$ and calculates the following
Deduce $\;ID_{SN}^{++}=\;ID_{SN}^{+}\;∥\;l*\;∥\;ID_{HN}^{+}$ from Formula (2), $ID_{FN}^{++}=ID_{FN}^{+}\;∥l*\;∥ID_{HN}^{+}$ from Formula (3), $ID_{HN}^{++}=\;ID_{SN}^{++}\;∥\;l*\;∥ID_{HN}^{+}\;∥\;ID_{FN}^{++}\;$ from Formula (4).Calculate $SK^{+}=h\left(ID_{SN}^{++}\;∥h\left(l*\;∥K_{MS}\right)\;∥ID_{FN}^{++}∥\;ID_{HN}^{++}\right)$ Formula (1)Compute $SID_{u}^{**}=h\left(ID_{U}\;∥ID_{FN}^{++}∥\;ID_{HN}^{++}∥\;rand^{+}\right)$ Formula (9) and update $F_{\left(i+1\right)}^{new}=h\left(SID_{u}^{**}\;∥ID_{FN}^{++}∥rand^{+}\right)$ from Formula (10), $G_{new}=h\left(SID_{u}^{**}\;∥SPW∥\;l*\;∥rand^{+}\right)$ from Formula (11). The above formulas ensure our scheme robustness towards jamming attack, due to the usage of old identities and keys in the generation of the new system parameters.Create security parameters to hide the new values:(15)$$J=rand\;⊕SID_{u}^{**},\;K=\;ID_{HN}^{++}⊕\;rand^{+}.$$

*HN* sends the parameters to the *FN* in the open channel $\left\{J,\;K,\;t_{j2}\right\}$.

**Step 3:***FN* → *SN* (*M*_3_ = {*J*, *K*, $t_{i3}$})

*FN* performs the following:Verify the time $\Delta t=t_{i3}-t_{j2}$ to prevent a replay attack.If $\Delta t=t_{i3}-t_{j2}>0$ then proceed, else halt the connection.

Upon receiving the parameters *FN* calculates the following:Deduce $SID_{u}^{**}=rand\;⊕$
J from Formula (15), $ID_{SN}^{++}=\;ID_{SN}^{+}\;∥\;l*\;∥\;ID_{HN}^{+}$ from Formula (2), $ID_{FN}^{++}=ID_{FN}^{+}\;∥l*\;∥ID_{HN}^{+}$ from Formula (3), $ID_{HN}^{++}=\;ID_{SN}^{++}\;∥\;l*\;∥ID_{HN}^{+}\;∥\;ID_{FN}^{++}\;$ from Formula (4) $rand^{+}=\;ID_{HN}^{++}⊕\;K$.Replace $F_{\left(i+1\right)}^{new}=h\left(SID_{u}^{**}\;∥ID_{FN}^{++}∥rand^{+}\right)$ from Formula (10), $G_{new}\;=h\left(SID_{u}^{**}\;∥SPW∥\;l*\;∥rand^{+}\right)$ from Formula (11), and add the new parameters to the SC {$F_{\left(i+1\right)}^{new},\;G_{new}\}$.

*FN* sends the parameters to the *SN* in the open channel $\left\{K,\;t_{i3}\right\}$.

**Step 4:** → *SN* (*M*_4_ = {*K*$,\;t_{i3}$})

Upon receiving the parameters *SN* calculates the following:Verify the time $\Delta t=t_{S4}-t_{i3}$ to prevent a replay attack.If $\Delta t=t_{s4}-t_{i3}>0$ then proceed, else terminate the session.Deduce $ID_{SN}^{++}=\;ID_{SN}^{+}\;∥\;l*\;∥\;ID_{HN}^{+}$ from Formula (2), $ID_{FN}^{++}=ID_{FN}^{+}\;∥l*\;∥ID_{HN}^{+}$ from Formula (3), $ID_{HN}^{++}=\;ID_{SN}^{++}\;∥\;l*\;∥ID_{HN}^{+}\;∥\;ID_{FN}^{++}\;$ from Formula (4).$rand^{+}=\;ID_{HN}^{++}⊕\;K$, then replace $SK^{+}=h\left(ID_{SN}^{++}\;∥h\left(l*\;∥K_{MS}\right)\;∥ID_{FN}^{++}∥\;ID_{HN}^{++}\right)$ Formula (1). Save the new parameters in the *SN* memory and establish the new session key.

### 5.4. P-II: Re-Authentication Phase

After an effective authentication session, the user is qualified to approach the system services. The authentic client might want to reach to some facilities throughout the day before night-time. Furthermore, it is timewasting, and un-efficient to compute all the values of the updated authentication session for the verified client. Hence, it necessitates the concept of re-authentication to improve the scheme efficiency, as shown in Figure 4. The stages of the re-authentication are as follows:The user enters to his/her account to approach some data from the smartphone.**Step 1:***FN* → *HN* (*M*_1_ = ($F_{\left(i_{+1}\right)},\;A,\;t_{i1}$))The *FN* calls the last $SID_{U},PW_{U}$ before the night-time to confirm the *FN* to the GW.Imprint the biometric BIO.Calculate $Gen\left(BIO\right)=\langle \delta_{i}|\tau_{i}\rangle \;$ from Formula (7) and $SPW=h\left(BIO\;∥ID_{FN}^{+}∥\;\delta_{i}\right)\;$ in Formula (8).Compute $G^{*}=h\left(SID_{U}\;∥SPW∥\;l*\;∥rand\right)$ from Formula (11).Check $G^{*}=?G$ if no abort, and if yes continue.Generate new time stamp $t_{i1}$ and calculate
(16)$$A_{i}=\;rand^{*}\;⊕\;SK⊕\;ID_{SN}^{+}⊕\;ID_{FN}^{+}.$$Send the following parameters **{**$F_{\left(i+1\right)},A_{i},\;t_{i1}$**}** to the *HN* for authentication.**Step 2:***HN* → *FN* (*M*_3_ = {$L,\;O,t_{j2}$})Verify the time $\Delta t=t_{j2}-t_{i1}$, to prevent replay attack.If $\Delta t=t_{j2}-t_{i1}>0$ then remain, else cancel the session.*HN* checks if $F_{\left(i+1\right)}$ was generated throughout the past 12 h.If yes, then *HN* gets the newest random nonce and computes the fresh key $A_{i}^{*}=\;rand^{*}\;⊕\;SK⊕\;ID_{SN}^{+}$
$⊕\;ID_{FN}^{+}$ from Formula (16).Check if $\;A_{i}^{*}=?\;A_{i}$ if equal, then proceed.Produce a new random nonce for the new connection *rand* (i), where i ={i+1, i+2, i+3, i + n |n: the number of new sessions}, then compute the following to replace the SC old parameters with the new ones: $F_{\left(i+1\right)}^{new}=h\left(SID_{U}\;∥ID_{FN}^{++}∥rand_{\left(i+1\right)}∥rand^{*}\right)$ from Formula (10), $G_{new}=h\left(SID_{U}\;∥SPW∥\;l*\;∥rand_{\left(i+1\right)}\right)$ from Formula (11).Produce recent time $t_{j2}\;$ and confirm the *HN* response.
(17)$$L=ID_{HN}^{+}⊕ID_{SN}^{++}\;⊕\;G_{new}$$
(18)$$O=F_{\left(i+1\right)}^{new}\;⊕\;rand^{*}\;$$Change the tuple values with the new ones $\left\{F_{\left(i+1\right)}^{new},\;G_{new}\right\}$, and save them to the SC.Send the following parameters {$L,\;O,\;t_{j2}\}$ to *FN*.**Step 3:** → *FN* (*M*_4_ = {$L,\;O,\;t_{j2}$})Check the time validity $\;\Delta t=t_{i3}-t_{j2}$. If $\Delta t=t_{i3}-t_{j2}>0$ then proceed, else halt the connection.Compute the following $G_{new}=ID_{HN}^{+}⊕ID_{SN}^{++}\;⊕\;L$ from Formula (17), $F_{\left(i+1\right)}^{new}=O\;⊕\;rand^{*}\;$ from Formula (18).Change the parameters with the fresh ones $\left\{F_{\left(i+1\right)}^{new},\;G_{new}\right\}$, and save them to the SC.

A protected connection can be initiated between *FN* and GW.

### 5.5. Secure Node Addition

SM adds new nodes to the system and performs the following:

**Step 1:** User sends a request to add new $SN_{i}^{new}$.

The client needs to normally log into the session with his/her credentials, inserts SC, enters $ID_{u}$ and $PW_{U}$, imprints the biometric BIO, and generates time stamp $t_{i1}.$
After a successful log in the user generates secret value for request validation.(19)$$M_{1}=h\left({}^{\prime }add\;node^{\prime },\;E,SID_{U}\right)$$Send this message to SM for node addition.

**Step 2:** SM receives the request of the user to create new $SN_{i}^{new}$.
SM checks the time validity $\;\Delta t=t_{j2}-t_{i1}$.If $\Delta t=t_{j2}-t_{i1}>0$ then proceed, else halt the session.SM opens the message to generate the new identity for the sensor $ID_{SN}^{new}$ and calculates $ID_{SN}^{new+}=ID_{SN}^{new}\;∥\;l*\;∥ID_{HN}$ from Formula (2) and make a new secret parameter $E=h\left(ID_{SN}^{new}\;∥h\left(l*\;∥ID_{HN}^{+}\right)\right)$ from Formula (5).The newly generated values are shared securely with the user and saved into the device’s memories.SM broadcast the new identity to all the communicating nodes for future access.

### 5.6. Secure User Revocation

In the below steps, SM revokes the user card from the system to add new one and performs the following:

**Step 1:** The user sends a request to remove his/her previous card and adds a new $SC_{NEW}\;$ to the system.

The user needs to normally log in to the session with his/her credentials, inserts SC, enters $SID_{u}$, $PW_{U}$, imprints the biometric BIO, and generates time stamp $t_{i1}.$
After a successful log in the user generates secret value for the request validation.(20)$$M_{1}=h\left({}^{\prime }revocation\;\&\;replace^{\prime },\;E,SID_{U},PW_{U}\;\right)$$Send this message to SM for card/mobile revocation.

**Step 2:** SM receives the user request to revoke from the system.
SM checks the time validity $\;\Delta t=t_{j2}-t_{i1}$.If $\Delta t=t_{j2}-t_{i1}>0$ then continue, else abort the session.SM checks the secret parameters and the request of the user via $E=h\left(ID_{SN}\;∥h\left(l*\;∥ID_{HN}^{+}\right)\right)$ from Formula (5).Send a secure link to the user to open, add his/her new $ID_{U}^{new}$, $SPW_{U}^{new}$, and $BIO_{new}$.Compute the following:(21)$$Gen\left(BIO_{new}\right)=\langle \delta_{i}|\tau_{i}\rangle$$
(22)$$SPW_{U}^{new}=h\left(BIO_{new}\;∥ID_{FN}^{+}∥\;\delta_{i}\right)$$The user sends the new parameter securely over the secure one-time link to the SM.SM receives the request, generates new random value $rand_{new}$, and computes the following:(23)$$SID_{U}^{new}=h\left(ID_{U}^{new}\;∥ID_{FN}^{+}∥\;ID_{HN}^{+}∥\;rand_{new}\right)$$
(24)$$F_{\left(i+1\right)}=h\left(SID_{U}^{new}\;∥ID_{FN}^{+}∥\;rand_{new}\right)$$
(25)$$G_{new}=h\left(SID_{U}^{new}\;∥SPW∥\;l*\;∥rand_{new}\right)$$
(26)$$H_{new}=h\left(rand_{new}\;∥ID_{FN}^{+}∥\;ID_{HN}^{+}\;∥ID_{SN}^{+}\right)$$Add the new values to the smart card $SC_{NEW}$.User receives the new smart card $SC_{NEW}$, replaces the new parameters $\left\{\;F_{\left(i+1\right)},\;G_{new},\;H_{new}\right\}$, and retrieves the random number from $\;H_{new}$, then deletes it from the new smart card in a secure channel. The new set of parameters will be $\left\{\;F_{\left(i+1\right)},\;G_{new}\;\right\}$.

### 5.7. Secure Data Transmission via Blockchain

In [12] scheme, there is no defined strategy to protect the stored data for retrieval or other usages after successful authentication. Since most of the WHSN structures are based on main centralized data storage that is accessible by the assigned doctor. So, this could put patient information in danger due to this source of error. Whereas the blockchain adds-up the data to blocks and splits them. Therefore, the integrity of the data is kept, each transaction is encrypted. Access control policies guarantee privacy [31]. Several methods were proposed to aid the purpose smart contract establishing along with patient identity tracking. In the case of government authorities who want to evaluate a medical facility service or measure the spread of a disease, the authorities need to have access to all the citizens’ information.

We adopted [32] method who proposed a Hyperledger blockchain which supports consensus algorithms that only permit the authenticated patients, and communications, and only accept reserved as well as confidential transactions. The Hyperledger blockchain consists of the transaction log that tackles all the changes made to the connections and changes the value of the world state. The blocks are built by a collection of transactions sent to the evaluator peer to simulate it, vote on it, and approve it. The structure of communication, electronic contracts, access policies are stored in the business network that the user can interact with from a mobile application connected to a server, where all the communication are encrypted by hashing to be able to access the blockchain for data storage and retrieval.

Another method is discussed by [21] that utilized the blockchain technology to store the individual data safely in the cloud. The sensor nodes contain some data that needs to be stored in the gateway safely for another retrieval or processing. The sensor sends encrypted data with the shared key to the foreign network along with the current timestamp. The foreign network node checks the timeliness and decrypts the data to get the information, then encrypts the data again with its pre-shared key to be sent to the hub node. The hub node decodes the info and checks the timestamp for validity to start building a data block. The block is added to the blockchain when all the communicated entities agreed upon the block contents in peer to peer cloud server network. After successfully gathering a group of valid data, the hub node starts to build transaction values and adds them together in one block to enable the system manager to create a blockchain of data for storage, deletion, update, and retrieve. The proposed method suggested the usage of cryptographic hash to encode the transmitted blocks and compute the ‘‘Merkle tree root’’ (MR) for the tree building. MR is a technique used in cryptocurrency to assure the data integrity in a peer-to-peer network structure. All the block information such as block owner and block payload are computed with the current block hash (CBHash). The hub node embeds the hashed identity of the user and sends the block of data to the system manager which uses ‘‘Ripple Protocol Consensus Algorithm (RPCA)’’ [33] for node verification and addition. Suppose that a user wants to access some data from a specific block, the user has to log in successfully to the connected hub node. So, as the hub node uses the user key that matches the user identity from the block, performs a hash function on data, decrypts the encrypted data to extract the hashes values and compare them with the computed hash for integrity check. Then, the hub node transmits the data to the user and the user decrypts the data with his/her key to retrieve the information from the block, as depicted in Figure 5.

## 6. Security Analysis

In this section, we discussed informal security analysis to analyze our scheme robustness against attacks in Section 6.2. In Section 6.3, we conducted a mathematical proof with BAN logic to confirm our structure mutual authentication and key agreement. Also, we simulate our model with Tamarin interactive tool to prove that our scheme is secure against session hijacking, replay attack, and attains perfect forward secrecy in Section 6.3.

### 6.1. Informal Security Evaluation

The below list indicated our system qualities.

#### 6.1.1. Mutual Authentication

Mutual authentication ensures that all communicating objects authenticate each other at the same time. In our protocol, we conducted the mutual authentication phase and all interactions between *FN* and *HN* in the authentication phase, and we conducted BAN logic formal proof along with simulation in Tamarin protocol to prove the mutual authentication. Thence, our scheme accomplishes mutual authentication because *HN* checks both user identity in the formula $SID_{u}^{*}=\;SID_{U}$ along with the sensor node identity in $ID_{SN}^{+}\;?=\;ID_{SN}^{+}$ before calculating the secret parameter. So, the mutual authentication is achieved in our scheme.

#### 6.1.2. Offline/Online Secret Shared Key Guessing

Regarding DY model, the intruder i can obtain all the parameters saved in the smart card, phone, *FN*, *SN*, and *HN*. Also, i can guess the perfect combination between username and password without the need to have SC or user mobile phone. Many elements protect our scheme from the attacks such as secret parameter l∗, the fresh biometric of the user BIO, the random values $rand^{*},\;\mathrm{and}\;rand^{+}$ that are checked observing their freshness, the secret parameters between nodes $F_{i1}$ and E, and the one-way hash function. Therefore, our scheme is robust against i shared secret key guessing in the online mode or offline mode.

#### 6.1.3. Nodes Anonymity

In the initialization phase, we masked all the important communicating objects identities with random values, and secret parameter. We concealed the *SN*, *FN*, *HN*, $ID_{U}$, nodes identities, and biometric BIO in Formulas (2)–(4), (8), and (9), respectively. Therefore, the intruder i cannot trace where the data came from and where it goes because of the anonymity and dynamicity of the connected objects’ identities. Moreover, the intruder i cannot guess the real identity of the user from $SID_{U}$ because it is protected by the power of hash function and a random number. Also, the biometric of the user is a unique value and it is hashed with the threshold value which stop any kind of guessing to this parameter. As a result, both protocols are holding the anonymity feature.

#### 6.1.4. Brute Force Attack

Intruder i can run a brute force attack on any identity, key or security parameters and can successfully know the correct parameters. Although our scheme makes it hard for the intruder i to guess the secure value correctly in polynomial time because we are implementing SHA-2 group of keys with size 224 bit, so by calculating the run-time of our key with one-way hash which is $\;2^{224}$. The intruder i cannot perform a brute force attack on our scheme due to the hash key size. The system’s key size fits our authentication procedure. However, it can be raised when necessary to attain security preliminaries in the future.

#### 6.1.5. Stolen Smartcard Attack

In [34] scheme, they did not specify a method to prevent brute force attacks in case of the lost smart card. Their paper did not mention the concept of encrypting and locking smartcard data information with user biometric or password. Therefore, we suggest in the cryptanalysis to reduce the number of parameters, random number checking, as well as smart card blocking policy after three times error in entering the authentication biometric, and password. Moreover, encrypting and locking the card information with a password, and user biometric at each time to authenticate the user to the smartcard will guarantee the tamper-resistant feature when the card is lost. Thus, our scheme prevents stolen smartcard attack.

#### 6.1.6. Replay Attack

All the communications between nodes during the authentication phase are protected by time stamp such that the communicating nodes generate new timestamp in each new parameter set creation. In the first communication between *FN* and *HN*, *FN* generates $t_{i1}$ and computes $A=\;rand^{*}\;⊕\;F_{i1}⊕ID_{SN}^{+}$ and $B=\;rand^{*}\;⊕SK$ for secret key as well as secret parameter confusion. In the second communication between *HN* and *FN*, *HN* generates $t_{j2}$ before updating the security parameters along with masking values. Therefore, Intruder i will not be able to crack the real session key or the hidden arguments in a valid time during successful communication.

#### 6.1.7. Integrity

Our scheme achieves integrity because all the security parameters, smart card parameters, biometric identity, and keys are protected with the one-way hash function. Moreover, the shadow identity of the user smart card is guarded with the formula $SID_{U}^{*}=h\left(F_{\left(i+1\right)}\;∥ID_{FN}^{+}∥\;rand\right)$, and the secret session key is protected by the formula $SK=h\left(ID_{SN}\;∥h\left(l*\;∥K_{MS}\right)\;∥\;ID_{FN}∥\;ID_{HN}\right)$. So, integrity is held in our both proposed protocols withal anonymity and un-traceability.

#### 6.1.8. Node Impersonation

Intruder i can compromise one communicating node and get its correct identity such as $ID_{FN}$ the real identity of the user stored in the smartphone memory. Although that the SC password SPW along with session key SK is protected by one-way hash function h(.)  along with the biometric, valid random number and secret parameter l∗. Besides, the intruder i is not able to compromise any other secret value or credential of the same communicating node or other nodes such as *HN* or *SN*. Subsequently, the proposed protocols are robust against any impersonation attack.

#### 6.1.9. Session Hijacking Attack

Intruder i can freely hijack any passing message in the public insecure medium. Also, the intruder i can hijack all the parameters sent among the communicated entities, collect them, and process them differently to elude the system security. Our security parameters are transmitted in the public medium are as few as possible, so the attacker will not get any useful information from collecting and intercepting the transmitted parameters between channels. The identity of the user is shadowed and protected by a one-way hash function ($F_{\left(i+1\right)},\;A,\;B,\;t_{i1}$) and ($\;J,\;K,\;t_{j2}$) where the attacker cannot guess the hidden parameters from the transmitted tuples. So, our proposed protocols are secure towards the session hijacking attack.

#### 6.1.10. Collision Attack

Intruder i goes for many permutations to crack the cryptographic hash and recovers arguments. This malicious act is useless in the proposed techniques since it is difficult to obtain two distinct messages that encompass the equal value in hash function h(m1) = h(m2). Thence, the robust hash function should stop collision [35]. So, in line with [17] the SHA- 2 cryptographic hash operation with size of the keys: 224 bit, 256 bit, and 384 bit, respectively, is protected versus collision attack.

#### 6.1.11. Scalability

Scalability is maintained when the growing of the system does not quite affect the performance of the system by increasing or decreasing a sensor or unit. In the case of fresh component adding or illegitimate component detection, our scheme is scalable by registering each user valid card, a sensor with unique security parameters, and IDs. Consequently, GW only permits the reliable nodes to make the connection and removes illegal nodes or cards in any future connection. Also, as per [36], to achieve scalability in the scheme, we should reduce the computation complexity for WHSN participating parties. Therefore, the core objective of this work is to boost the performance of [12] so we had accomplished our objective by decreasing cost of telecommunications.

#### 6.1.12. Forward/Backward Secrecy

Forward secrecy evading is the capability of the intruder i to anticipate the potential key pair. Whereas backward secrecy happens when the attacker gathers as many previous keys as necessary to infer the former keys. The session keys are dynamic and secured in our schemes by several parameters such as random nonce, new foreign network identification, hidden value, and the timestamp. Thus, even though the intruder i correctly identified the keys, due to the complicated parametric method, he/she is unable to predict the future keys or breach the prior keys. In addition, the intruder i needs to correctly predict the following: $ID_{FN}^{++}=ID_{FN}^{+}\;∥l*\;∥ID_{HN}^{+}$, $ID_{HN}^{++}=\;ID_{SN}^{++}\;∥\;l*\;∥ID_{HN}^{+}\;∥\;ID_{FN}^{++}$, $SK^{+}=h\left(ID_{SN}^{++}\;∥h\left(l*\;∥K_{MS}\right)\;∥ID_{FN}^{++}∥\;ID_{HN}^{++}\right)$, $SID_{u}^{**}=h(ID_{U}\;∥ID_{FN}^{++}∥\;ID_{HN}^{++}∥\;rand^{+}\;$ to be able to disclose all the hidden data in the session. Consequently, optimal forward and backward secrecy is accomplished by our protocols.

#### 6.1.13. Jamming Attack

Intruder i tries to disrupt the authentication process by generating a jamming signal to prevent the exchanging of some parameters during the communication. In our scheme, we enabled *FN* and *HN* to generate two random numbers that aided the key establishment. The last generated key and identities are used in the creation of the new parameters. So, the Intruder i needs to be aware of the formed session keys, identities, and random values to generate a successful jamming attack. Also, our scheme is protected by a timestamp that prevents the attacker from using old parameters after a long-time passage because the scheme will halt the expired session.

### 6.2. Ban Logic Proof

In this section, a formal proof with BAN logic method is conducted to prove our scheme mutual authentication and key agreement for P-I:

#### 6.2.1. Essential Symbolization

The following covers the over-all fundamental representation for BAN logic to be employed in protocols P-I and P-II, see Table 3:

#### 6.2.2. P-I: Goals

The goals to be achieved by P-I are stated below:


$\mathrm{Goal}\;1\;\to FN\;|\equiv \left(FN\;\overset{SK}{↔}HN\right)$



Goal 2 →HN |≡(FN ↔SKHN)



$\mathrm{Goal}\;3\;\to FN\;\left|\equiv HN\;\right|\equiv \left(FN\;\overset{SK}{↔}HN\right)$



$\mathrm{Goal}\;4\;\to HN\;\left|\equiv FN\;\right|\equiv \left(FN\;\overset{SK}{↔}HN\right)$


##### P-I: Idealized Form

Below, we mentioned the ideal messages forms on the P-I:


$Msg1:\;FN\;\to HN\;{\left(ID_{U},\;SID_{U},\;rand^{*},\;t_{i1}\right)}_{rand^{*}}$



$Msg2:HN\;\to FN\;{\left(SID_{U},\;ID_{SN}^{++},\;rand^{*},\;rand^{+},\;t_{j2}\right)}_{rand^{+}}$



$Msg3:FN\;\to SN\;{\left(SID_{U},\;ID_{SN}^{++},\;rand^{*},\;rand_{\left(i+1\right)},\;t_{I3}\right)}_{rand^{+}}$



$Msg4:\;\to SN\;{\left(SID_{U},\;rand^{*},rand_{\left(i+1\right)},\;t_{S4}\right)}_{rand^{+}}$


##### P-I: Assumptions

In the following, we explained the assumption of P_I:


$A1:HN\;|\equiv \#\left(t_{i1}\right)$



$A2:FN\;|\equiv \#\left(t_{j2}\right)$



$A3:SN\;|\equiv \#\left(t_{i3}\right)$



$A4:FN\;|\equiv \#\left(t_{S4}\right)$



$A5:HN\;|\equiv \#\left(FN\;\overset{rand^{*}}{↔}HN\right)$



$A6:FN\;|\equiv \#\left(FN\;\overset{rand^{*}}{↔}HN\right)$



$A7:SN\;|\equiv \#\left(FN\;\overset{rand^{+}}{↔}SN\right)$



$A8:FN\;|\equiv \#\left(FN\;\overset{rand^{+}}{↔}SN\right)$



$A9:FN\;|\equiv HN\;⟹\#\left(FN\;\overset{SK}{↔}HN\right)$



$A10:HN\;|\equiv FN\;⟹\#\left(FN\;\overset{SK}{↔}HN\right)$



$A11:FN\;|\equiv SN\;⟹\#\left(FN\;{\overset{SK}{↔}}^{+}SN\right)$



$A12:SN\;|\equiv FN\;⟹\#\left(FN\;{\overset{SK}{↔}}^{+}SN\right)$


##### P-I: BAN Logic Proof

The BAN logic proof is processed as follows.

**Step 1**: according to Msg1 we get:$$HN\leftarrow {\left(ID_{U},\;SID_{U},\;rand^{*},t_{i1}\right)}_{rand^{*}}$$

**Step 2:** using Step 1 with “message meaning rule”:$$HN\left|\equiv FN\;\right|\sim \;{\left(ID_{U},\;SID_{U},\;rand^{*},t_{i1}\right)}_{rand^{*}}$$

**Step 3:** using Step 2 and A1 with “freshness rule”:$$HN|\equiv \#\;{\left(ID_{U},\;SID_{U},\;rand^{*},t_{i1}\right)}_{rand^{*}}$$

**Step 4:** using Step 2 and Step 3 with “random nonce verification rule”:$$HN\left|\equiv FN\;\right|\equiv {\left(ID_{U},\;SID_{U},\;rand^{*},t_{i1}\right)}_{rand^{*}}$$

**Step 5:** according to Msg2 we get:$$FN\leftarrow \;{\left(SID_{U},ID_{SN}^{++},\;rand^{*},rand^{+},t_{j2}\right)}_{rand^{+}}$$

**Step 6:** using Step 5 and A6 “message meaning rule”:$$FN\leftarrow \;{\left(SID_{U},ID_{SN}^{++},\;rand^{*},rand^{+},t_{j2}\right)}_{rand^{+}}$$

**Step 7:** using Step 6 and A3 “freshness rule”:$$FN|\equiv \#\;{\left(SID_{U},ID_{SN}^{++},\;rand^{*},rand^{+},t_{j2}\right)}_{rand^{+}}$$

**Step 8:** using Step 6 and Step 7 “random nonce verification rule”:$$FN\left|\equiv HN\right|\equiv {\left(SID_{U},ID_{SN}^{++},\;rand^{*},rand^{+},t_{j2}\right)}_{rand^{+}}$$

**Step 9:** according to Msg3 we get:$$SN\leftarrow {\left(SID_{U},ID_{SN}^{++},\;rand^{*},rand^{+},t_{j3}\right)}_{rand^{+}}$$

**Step 10:** using Step 9 and A5 with “message meaning rule”:$$FN\left|\equiv SN\right|\sim \;{\left(SID_{U},ID_{SN}^{++},\;rand^{*},rand^{+},t_{j3}\right)}_{rand^{+}}$$

**Step 11:** using Step 10 and A2 with “freshness rule”:$$FN|\equiv \#{\left(SID_{U},ID_{SN}^{++},\;rand^{*},rand^{+},t_{j3}\right)}_{rand^{+}}$$

**Step 12:** using Step 10 and Step 11 with “random nonce verification rule”:$$FN\left|\equiv SN\;\right|\equiv {\left(SID_{U},ID_{SN}^{++},\;rand^{*},rand^{+},t_{j3}\right)}_{rand^{+}}$$

**Step 13**: from Msg4 we get:$$SN\;\leftarrow {\left(SID_{U},\;rand^{*},rand^{+},t_{S4}\right)}_{rand^{*}}$$

**Step 14:** using Step 13 and A7 with “message meaning rule”:$$SN\left|\equiv FN\;\right|\sim {\left(SID_{U},\;rand^{*},rand^{+},t_{S4}\right)}_{rand^{*}}$$

**Step 15:** using Step 14 and A4 with “freshness rule”:$$SN\left|\equiv FN\;\right|\;\#{\left(SID_{U},\;rand^{*},rand^{+},t_{S4}\right)}_{rand^{*}}$$

**Step 16:** using Step 14 and Step 15 with “random nonce verification rule”:$$SN\left|\equiv FN\;\right|\;\#{\left(SID_{U},\;rand^{*},rand^{+},t_{S4}\right)}_{rand^{*}}$$

**Step 17:** because $SPW=h\left(rand\;∥rand^{+}\right)$ we get Step 12 and Step 6 (Goal 3)
$$FN\left|\equiv HN\;\right|\equiv \left(FN\;{\overset{SK}{↔}}^{+}HN\right)$$

**Step 18:** because $SPW=h\left(rand\;∥rand^{+}\right)$, according to Step 4 and Step 8 (Goal 4)
$$HN\left|\equiv FN\;\right|\equiv \left(FN\;\overset{{\overset{SK}{↔}}^{+}}{↔}\;SN\right)$$

**Step 19:** from A9 and Step 17 (Goal 1)
$$FN\;|\equiv \left(FN\;\overset{SK}{↔}\;HN\right)$$

**Step 20:** from A10 and Step 18 (Goal 2)
$$HN\;|\equiv \left(FN\;\overset{SK}{↔}\;HN\right)$$

In this section, a formal proof with BAN logic method is conducted to prove our scheme mutual authentication and key agreement for P-II:

#### 6.2.3. P-II: Goals

The ideal goals to be achieved by P-II are stated as follows:


$\mathrm{Goal}\;1\;\to FN\;|\equiv \left(FN\;\overset{SK}{↔}HN\right)$



Goal 2 →HN |≡(FN ↔SKHN)



$\mathrm{Goal}\;3\;\to FN\;\left|\equiv HN\;\right|\equiv \left(FN\;\overset{SK^{+}}{↔}HN\right)$



$\mathrm{Goal}\;4\;\to HN\;\left|\equiv FN\;\right|\equiv \left(FN\;\overset{SK^{+}}{↔}HN\right)$


##### P-II: Idealized Form

Below, we illustrated the idealized form of the message to be transmit between nodes in P-II:


$Msg1:\;FN\;\to HN\;{\left(ID_{U},\;SID_{U},\;rand^{*},\;t_{i1}\right)}_{rand^{*}}$



$Msg2:HN\;\to FN\;{\left(SID_{U},\;ID_{SN}^{++},\;rand^{*},\;rand^{+},\;t_{j2}\right)}_{rand_{\left(i+1\right)}}$



$Msg3:FN\;\to SN\;{\left(SID_{U},\;ID_{SN}^{++},\;rand^{*},\;rand^{+},\;t_{I3}\right)}_{rand_{\left(i+1\right)}}$



$Msg4:\;\to SN\;{\left(SID_{U},\;rand^{*},\;rand^{+},\;t_{S4}\right)}_{rand_{\left(i+1\right)}}$


##### P-II: Assumptions

Same assumption as before just change A5–A8:


$A5:HN\;|\equiv \#\left(FN\;\overset{rand_{\left(i+1\right)}}{↔}HN\right)$



$A6:FN\;|\equiv \#\left(FN\;\overset{rand_{\left(i+1\right)}}{↔}HN\right)$



$A7:SN\;|\equiv \#\left(FN\;\overset{rand_{\left(i+1\right)}}{↔}SN\right)$



$A8:FN\;|\equiv \#\left(FN\;\overset{rand_{\left(i+1\right)}}{↔}SN\right)$


##### P-II: BAN Logic Proof

The BAN logic proof then proceeds as below.

**Step 21**: According to Msg1 we get:$$HN\leftarrow {\left(ID_{U},\;SID_{U},\;rand^{*},t_{i1}\right)}_{rand^{*}}$$

**Step 22:** Using Step 21 with “message meaning rule”:$$HN\left|\equiv FN\;\right|\sim \;{\left(ID_{U},\;SID_{U},\;rand^{*},t_{i1}\right)}_{rand^{*}}$$

**Step 23:** Using Step 22 and A1 with “freshness rule”:$$HN|\equiv \#\;{\left(ID_{U},\;SID_{U},\;rand^{*},t_{i1}\right)}_{rand^{*}}$$

**Step 24:** Using Step 22 and Step 23 with “random nonce verification rule”:$$HN\left|\equiv FN\;\right|\equiv {\left(ID_{U},\;SID_{U},\;rand^{*},t_{i1}\right)}_{rand^{*}}$$

**Step 25:** According to Msg2 we get:$$FN\leftarrow \;{\left(SID_{U},ID_{SN}^{++},\;rand^{*},rand_{\left(i+1\right)},t_{j2}\right)}_{rand_{\left(i+1\right)}}$$

**Step 26:** Using Step 25 and A6 “message meaning rule”:$$FN\leftarrow \;{\left(SID_{U},ID_{SN}^{++},\;rand^{*},rand_{\left(i+1\right)},t_{j2}\right)}_{rand_{\left(i+1\right)}}$$

**Step 27:** Using Step 26 and A3 “freshness rule”:$$FN|\equiv \#\;{\left(SID_{U},ID_{SN}^{++},\;rand^{*},rand_{\left(i+1\right)},t_{j2}\right)}_{rand_{\left(i+1\right)}}$$

**Step 28:** Using Step 26 and Step 27 “random nonce verification rule”:$$FN\left|\equiv HN\right|\equiv {\left(SID_{U},ID_{SN}^{++},\;rand^{*},rand_{\left(i+1\right)},t_{j2}\right)}_{rand_{\left(i+1\right)}}$$

**Step 29:** According to Msg3 we get:$$SN\leftarrow {\left(SID_{U},ID_{SN}^{++},\;rand^{*},rand_{\left(i+1\right)},t_{j3}\right)}_{rand_{\left(i+1\right)}}$$

**Step 30:** Using Step 29 and A5 with “message meaning rule”:$$FN\left|\equiv SN\right|\sim \;{\left(SID_{U},ID_{SN}^{++},\;rand^{*},rand_{\left(i+1\right)},t_{j3}\right)}_{rand_{\left(i+1\right)}}$$

**Step 31:** Using Step 30 and A2 with “freshness rule”:$$FN|\equiv \#{\left(SID_{U},ID_{SN}^{++},\;rand^{*},rand_{\left(i+1\right)},t_{j3}\right)}_{rand_{\left(i+1\right)}}$$

**Step 32:** Using Step 30 and Step 31 with “random nonce verification rule”:$$FN\left|\equiv SN\;\right|\equiv {\left(SID_{U},ID_{SN}^{++},\;rand^{*},rand_{\left(i+1\right)},t_{j3}\right)}_{rand_{\left(i+1\right)}}$$

**Step 33**: From Msg4 we get:$$SN\;\leftarrow {\left(SID_{U},\;rand^{*},rand_{\left(i+1\right)},t_{S4}\right)}_{rand^{*}}$$

**Step 34:** According Step 33 and A7 with “message meaning rule”:$$SN\left|\equiv FN\;\right|\sim {\left(SID_{U},\;rand^{*},rand_{\left(i+1\right)},t_{S4}\right)}_{rand^{*}}$$

**Step 35:** From Step 34 and A4 with “freshness rule”:$$SN\left|\equiv FN\;\right|\;\#{\left(SID_{U},\;rand^{*},rand_{\left(i+1\right)},t_{S4}\right)}_{rand^{*}}$$

**Step 36:** Using Step 34 and Step 35 with “random nonce verification rule”:$$SN\left|\equiv FN\;\right|\;\#{\left(SID_{U},\;rand^{*},rand_{\left(i+1\right)},t_{S4}\right)}_{rand^{*}}$$

**Step 37:** Because $SPW=h\left(rand_{\left(i+1\right)}\;∥rand^{*}\right)$ we get Step 32 and Step 26 (Goal 3)
$$FN\left|\equiv HN\;\right|\equiv \left(FN\;{\overset{SK}{↔}}^{+}HN\right)$$

**Step 38:** Due $SPW=h\left(rand_{\left(i+1\right)}\;∥rand^{*}\right)$, according to Step 24 and Step 28 (Goal 4)
$$HN\left|\equiv FN\;\right|\equiv \left(FN\;\overset{{\overset{SK}{↔}}^{+}}{↔}\;SN\right)$$

**Step 39:** From A9 and Step 37 (Goal 1)
$$FN\;|\equiv \left(FN\;\overset{SK}{↔}\;HN\right)$$

**Step 40:** From A10 and Step 38 (Goal 2)
$$HN\;|\equiv \left(FN\;\overset{SK}{↔}\;HN\right)$$

### 6.3. Simulation with Tamarin Prover

We simulated our scheme with Tamarin prover [37] to prove our scheme robustness against session hijacking, replay attacks, perfect forward secrecy, and mutual authentication. It is a tool used for formal protocols validation and written in the Haskell language. The simulation was operated on MacBook Air, it ran on MacOS Catalina, with processor 1.8 GHz Dual-Core Intel Core i5, Memory 8 GB 1600 MHz DDR3, and Intel HD Graphics 6000 1536 MB. Also, we uploaded some tools to help our system to simulate the protocol which are graphviz version 2.44.1, maude tool, SAPIC tool, and sublime text to show colorful coding for the protocol syntax.

#### Haskell Specification

We specified our nodes in the communication model as *FN* (user), *HN* (gateway), and *SN* (sensor node) to be represented in the Tamarin environment with their specified interaction along with attack simulation to ensure the scheme validity and robustness against the simulated attacks.

In Figure 6, we showed how nodes, secret key, biometric, and smart card are registered by the SM over a secure channel. Then when the user received the smart card, the user registers the biometric BIO, the identity $SID_{U}$, and the password to form the following parameters $\left\{\;F_{\left(i+1\right)},\;G\;Gen\left(\;\right),\;Rep\left(\;\right),\;h\;\left(\;\right)\;\right\}$. The registered client inserts the smart card SC, and starts the authentication between the *FN* and *HN*.

We showed how the *FN* gets all the parameters from the user and calculates the masking parameters ($\;F_{\left(i+1\right)}\;,\;A,\;B,\;t_{i1}$) to be transmitted to the *HN*, as depicted in Figure 7.

Next, *HN* receives the authentication parameters and authenticates the user to start calculating the new set of parameters ($\;J,\;K,\;t_{j2}$) for the card secret data renewal as well as the nodes’ identities and secret keys, as depicted in Figure 8 and Figure 9.

We specified some lemmas to ensure our parameters and nodes secrecy in a matter of session hijacking and guarantee that our scheme holds perfect forward secrecy, as depicted in Figure 10.

Also, we specified other lemmas to prove our scheme parameters secrecy against replay attack, as depicted in Figure 11.

The below results describe that the scheme holds the property by highlighting the codes in green color. Our scheme fulfils the perfect forward secrecy, resistance against replay attack, and session hijacking attack in both *HN* and *FN*, as depicted in Figure 12 and Figure 13, respectively.

The below graphs illustrate, that our scheme fulfils the perfect forward secrecy, resistance against replay attack, and session hijacking attack in both *HN* and *FN* with absence of the red arrows in the figures, as depicted in both Figure 14 and Figure 15.

In the next section, we provided the performance analysis to the scheme to show its efficiency in comparison with other schemes.

## 7. Performance Analysis

In the following, we deliberated the efficiency of the proposed protocols regarding their computation, communication, and storage.

### 7.1. Computational Cost

In this section, the computation cost calculation is performed for the proposed protocols that employed a cryptographic hash which takes 0.00032 s along with a biometric reproduce operation that takes 0.0171 s based on the metrics in [38]. The computational cost of the proposed scheme is better than all other schemes in the foreign network side by 60% and 65% and *HN* side [17,19,22] with a 80% by using P-I and 85% by using P-II. Besides, P-I and P-II perform better than [12,16] in the foreign network side along with *HN* side with 5% and 15% reduction, respectively, as depicted in Figure 16 and Figure 17. Furthermore, we chose a hash function with a 224 bit key size to allow the foreign network to have an adequate level of security better than [12] which takes a 160 bit key size, and [19,22] which takes a 128 bit size key, (see Table 4, Table 5, Table 6 and Table 7).

### 7.2. Communication Overhead

We assumed the length of the hash function, keys, and security parameters = 224 bits, and the timestamp = 32 bits. Besides, our system contains four tuples in the foreign network side ($F_{\left(i+1\right)},\;A,\;B,\;t_{i1}$) that results in = 224 + 224 + 224 + 32 = 704 bit. Moreover, we have ($\;J,\;K,\;t_{j2}$) from *HN* to *FN* that results in = 224 + 224 + 32 = 480 bit. Those results demonstrate that our system has the least overhead in a GW side more than all schemes in the comparison [16,17,19,22], with more strength versus numerous attacks, as shown in Table 8.

### 7.3. Storage Overhead

We determined the storage cost of our work in contrast to [16,17,19,22] schemes to analyze the schemes’ capacities. Assuming that each function and parameter of the following have different storage bytes such that hash, ECC, AES symmetric, RSA asymmetric, parameters identifications, random number, and time are 20, 20, 20, 20, 4, 4, and 16 bytes, respectively, and the prime p in Ep (a, b) is 20 bytes. The suggested scheme requires storage for the stored arguments $\left\{\;F_{\left(i+1\right)},\;G_{new}\right\}$ that results in (20+20=40 bytes) for the smartcard, and rand requires 20 bytes for the gateway. The storage cost distinguishes our scheme from others because it is the lowest of all on the smart card side. Moreover, the number of stored security parameters in the proposed structure will provide better security among other schemes, as shown in Table 9 and Table 10.

From Table 6, we compared our proposed scheme computation cost to schemes, and we identified that our scheme performs better than all in both foreign network and hub node sides. Y. Park and Y. Park scheme [16] scheme contains 20 hashes, a fuzzy extractor, and 2 ECC point multiplications. C. Wang et al. scheme [17] requires 8 hashes, and 26 ECC point multiplications. Mo and Chen scheme [19] takes 22 hashes, a reproduction operation, 2 Modular exponentiations, and 1 symmetric encryption. Similarly, Ali et al. scheme [22] scheme needs 7 hashes, 3 ECC point multiplications, and a fuzzy extractor. Yu and Park scheme [12] takes 22 hashes, and a reproduction function. In comparison to the proposed scheme, our authentication protocol requires 12 hashes along with 1 reproduction function, and the re-authentication protocol requires 4 hashes and 1 reproduction function. This manifests that our scheme has a lower computational cost and low energy consumption.

## 8. Conclusions

WHSN is a modern trend that deals with significant information from the patients that must be protected. It received major attention from the information security developers and vendors, who put big efforts in increasing the guardedness of the WHSN system and speed up the performance. Therefore, we analyzed the latest schemes in the field and we found that [12] to be the most efficient and secure one. So, we cryptanalyze it and we discovered that the scheme needs enhancement to achieve both security and performance. Consequently, a three-factor authentication scheme based on the biometric, smart card, and password is proposed. The scheme was formally validated by BAN logic and simulated with Tamarin prover to confirm its security and mutual authentication. Moreover, the informal analysis proved that the above scheme achieved the suggested security requirements like, anonymity, offline/online shared secret guessing, *FN*-*SN* replay attack, brute force attack, *FN*/*HN* impersonation, integrity, session hijacking, eavesdropping attack, un-traceability, and collision attack. Finally, we conducted performance evaluation to compute our scheme efficiency and we found out that our scheme has better computation cost, communication cost, storage cost, and energy consumption than the related schemes. To conclude, the future direction of our research will employ blockchain technology in WHSN authentication in-depth and more attacks simulation in the proverif tool.

## Figures and Tables

**Figure 1 sensors-20-06860-f001:**
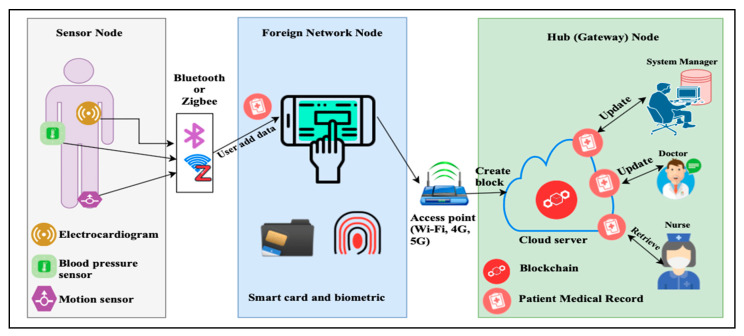
Network model for the Wireless Healthcare Sensor Network (WHSN) system.

**Figure 2 sensors-20-06860-f002:**
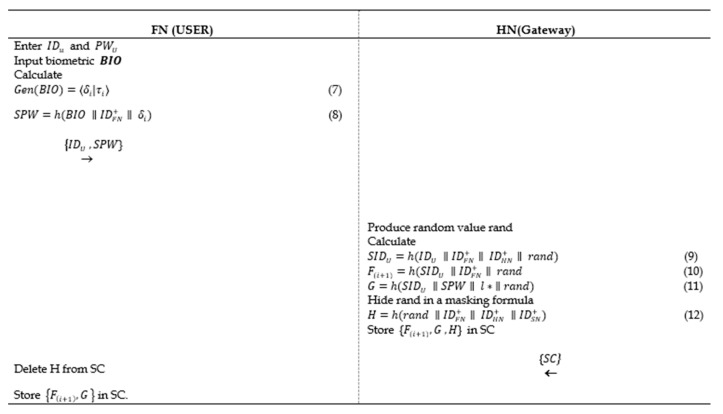
The proposed scheme registration phase.

**Figure 3 sensors-20-06860-f003:**
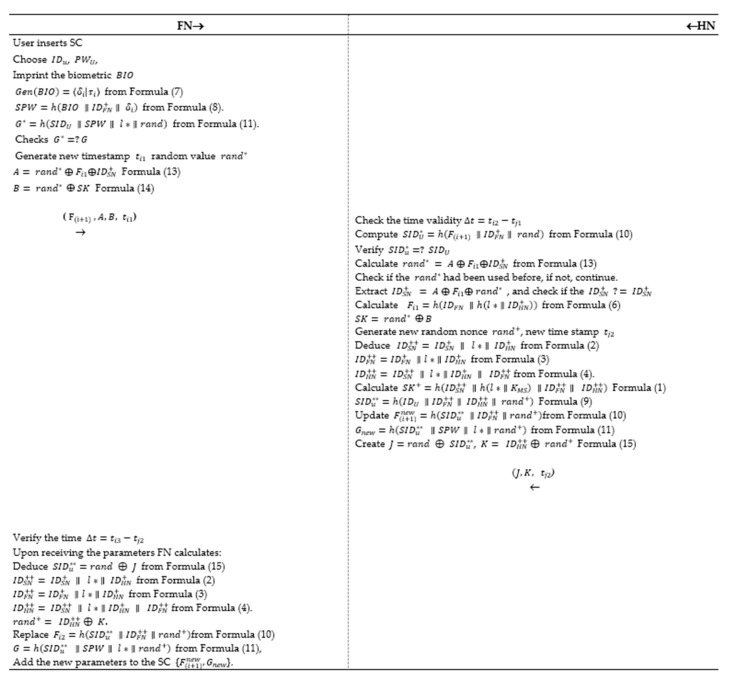
P-I: Three-factor authentication and key agreement protocol.

**Figure 4 sensors-20-06860-f004:**
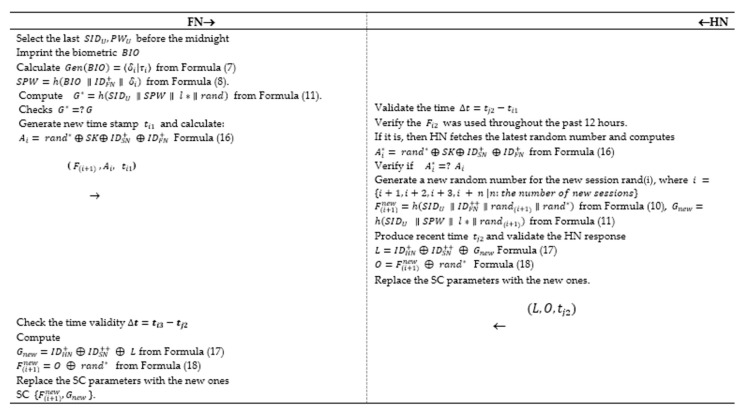
P-II: Three-factor re-authentication and key agreement protocol.

**Figure 5 sensors-20-06860-f005:**
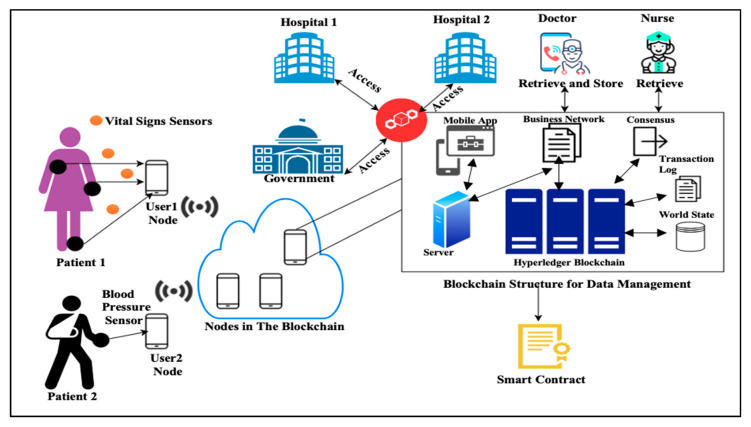
Medical record formation to the blockchain in the cloud.

**Figure 6 sensors-20-06860-f006:**
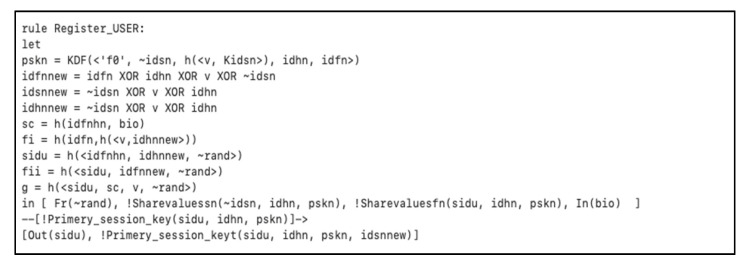
Initialization and registration phase for the user and other nodes.

**Figure 7 sensors-20-06860-f007:**
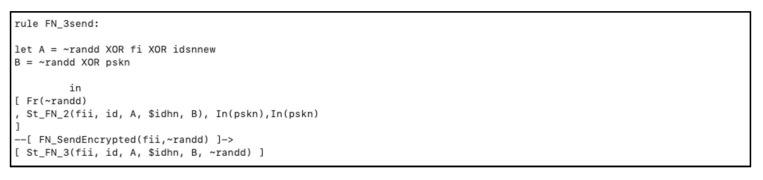
*FN* send authentication request to the *HN*.

**Figure 8 sensors-20-06860-f008:**
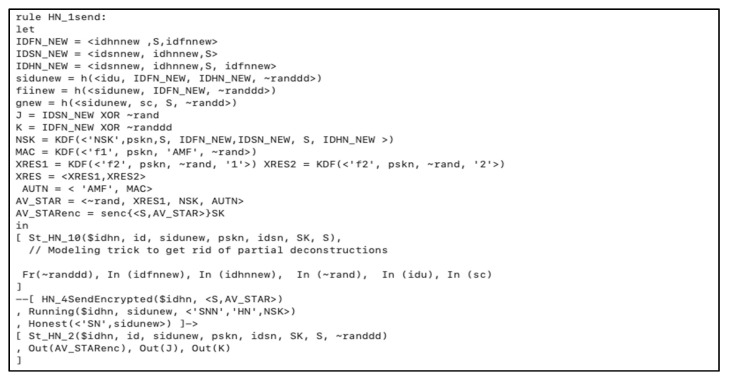
*HN*–*FN* authentication and new parameters creation.

**Figure 9 sensors-20-06860-f009:**
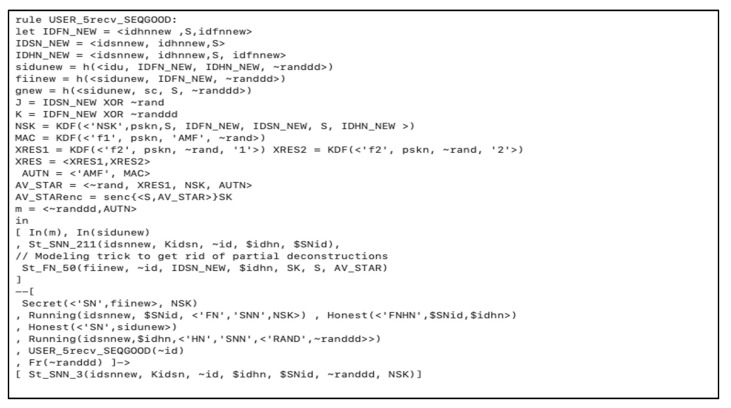
*FN* new parameters calculation and honesty verification.

**Figure 10 sensors-20-06860-f010:**
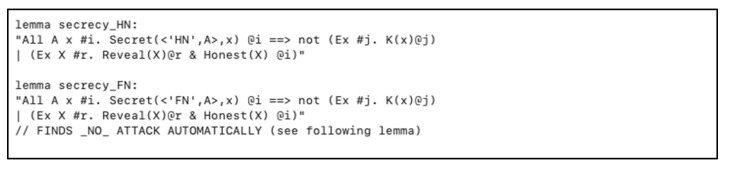
Session hijacking and perfect forward secrecy resistance lemmas in *HN* and *FN*.

**Figure 11 sensors-20-06860-f011:**
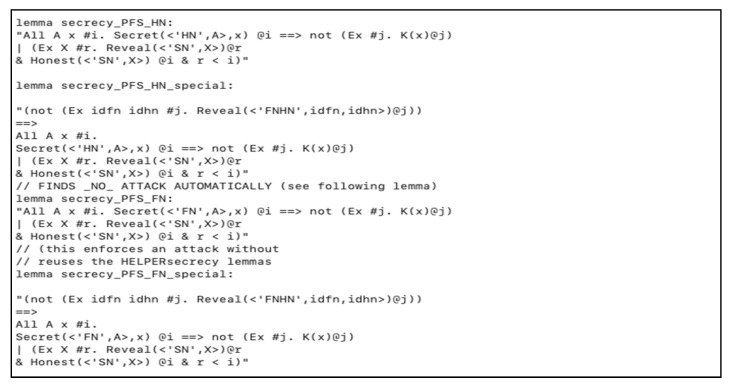
Replay attack resistance lemmas in *HN* and *FN*.

**Figure 12 sensors-20-06860-f012:**
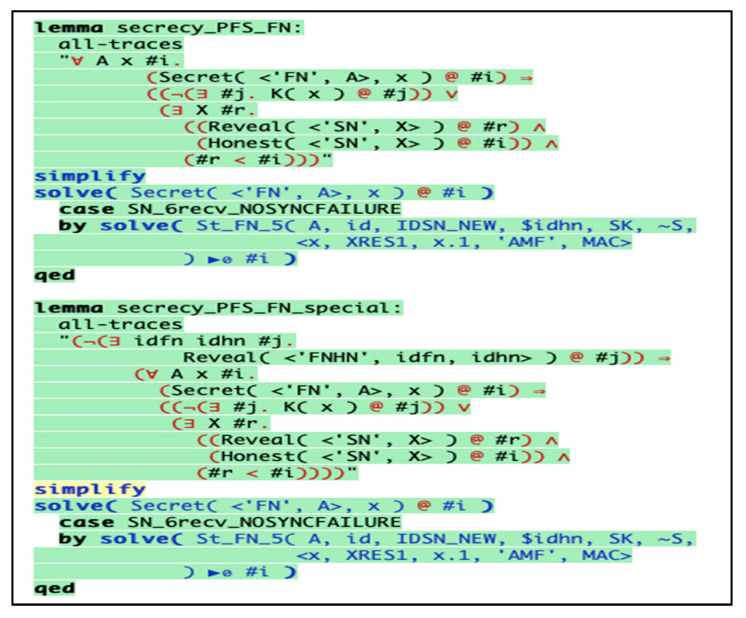
Replay attack resistance in *FN*.

**Figure 13 sensors-20-06860-f013:**
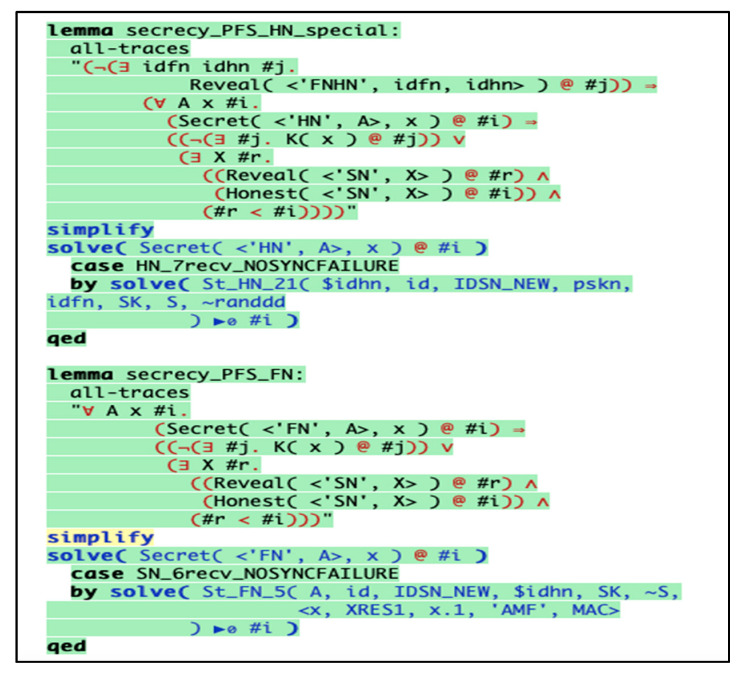
Replay attack resistance in *HN*.

**Figure 14 sensors-20-06860-f014:**
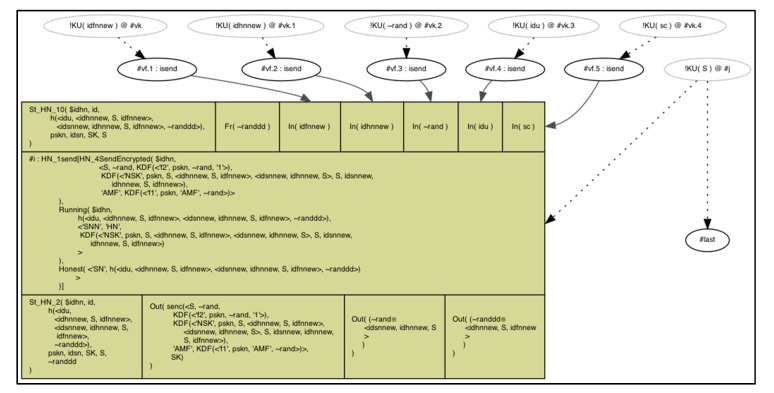
Secrecy of the scheme parameters during transmission on the public channel by *HN*.

**Figure 15 sensors-20-06860-f015:**
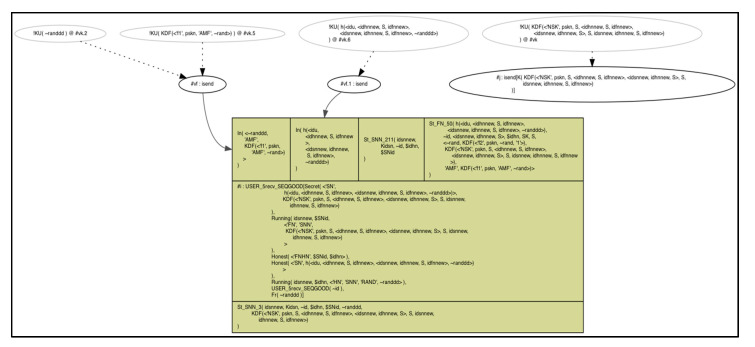
Secrecy of the scheme parameters during reception from the public channel by *FN*.

**Figure 16 sensors-20-06860-f016:**
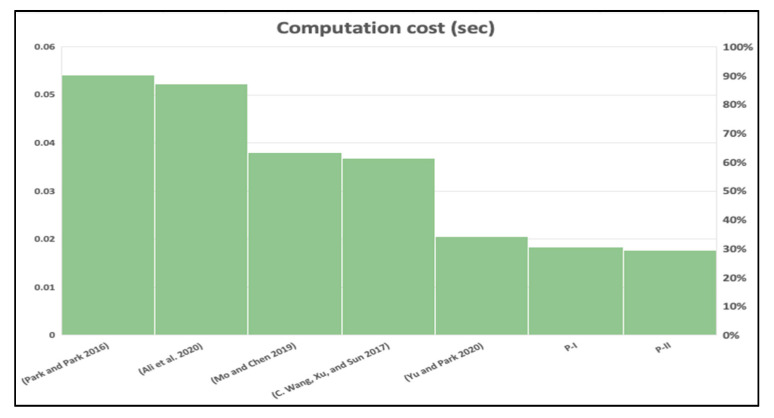
Computation time in foreign network node.

**Figure 17 sensors-20-06860-f017:**
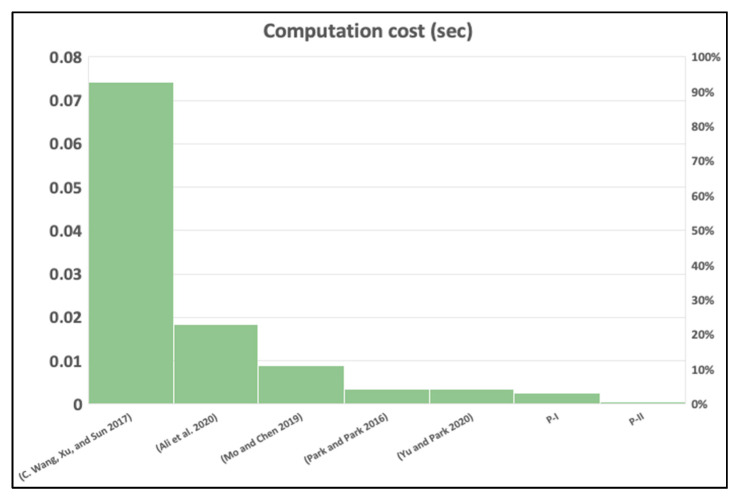
Computation time in hub node.

**Table 1 sensors-20-06860-t001:** Symbols used in Yu and Park protocol.

Notation	Description
$U_{i}$	User
GWN	Gateway node
$S_{j}$	Sensor node
$ID_{i}$	*Ui*’s identity
$PW_{i}$	*Ui*’s password
$SID_{j}$	*Sj*’s identity
$K_{GWN}$	Master key of *GWN*
$X_{pub}$	Public key of *GWN*
$x_{j}$	Secret key of *Sj*
$E,F_{p}$	Elliptic curve *E* defined on the finite field *Fp* with order *p*
G	A group for an elliptic curve
P	The generator of *G*
$E_{k},D_{k}$	Symmetric key encryption/decryption
SK	Session key
$T_{i}$	Timestamp
BIO	Biometric of *Ui*
h(.)	Hash function
⊕	XOR operation
∥	Concatenation operation

**Table 2 sensors-20-06860-t002:** Notations of the scheme.

Symbol	Description
SM	System controller (manager)
SN	Second-level node (sensor)
FN/ *(user)*	Foreign network node/user
HN *(gw)*	Gateway node (hub)
SC	Smart card
$ID_{u}$ $,\;PW_{U}$	User identity and password picked by the user
$SID_{U},\;SPW_{U},\;SID_{U}^{**}$	User shadow identity and shadow password/updated shadow identity
$ID_{SN},\;ID_{SN}^{+},$	Second-level node identity generated by SM/hidden ID_SN_ updated/changed ID_SN_ constantly.
$ID_{SN}^{++}$	
$ID_{FN},\;ID_{FN}^{+}$	Foreign network node identity (user identity) created by SM/masked ID_FN_
$ID_{FN}^{++}$	Updated Foreign network in every period or in every updated user identity
$ID_{HN},\;ID_{HN}^{+},\;ID_{HN}^{++}$	Gateway identity created by the system controller/hidden HN identity/updated HN identity
$t_{i},\;t_{j},\;t_{S}$	Recent time of the Foreign network and GW nodes
$K_{MS}$	Master secret key created by the controller pre-shared between FN and HN
SK	Session key computed by SM
$SK^{+}$ $,SK_{n}^{+}$	Renewed session key/updated symmetric key
$rand,\;rand^{*},rand^{+}$	Random nonce/renewed random nonce
BIO	Biometric of the user
$DB_{HN}$	Database of the hub node
$E,F_{i1},F_{\left(i+1\right)},A,B,J,K$	Verification parameters
⊕	XOR operator
h(.)	Cryptographic hash
∥	Concatenation operation

**Table 3 sensors-20-06860-t003:** Symbols used in the BAN (Burrows Abadi Nadeem) logic.

Symbols	Description
*M*|≡ *N*	*M* trusts *N*
*M*Δ*N*	*M* sees *N*
*M*| ∼ *N*	*M* once responded *N*
*M*|=〉 *N*	*M* governs *N*
#(*N*)	*N* is new
〈*N*〉*B*	*N* is merged with *B*
〈*N*〉*B*	*N* is encrypted by *B*
$N\;\overset{K\;}{\Leftrightarrow }\;$ *Q*	*K* is shared secret between *N* and *Q*
SK	Pre-shared key used in connection

**Table 4 sensors-20-06860-t004:** Rough estimated time for various schemes [38].

Notation	Description	Computation Time in Seconds
$T_{h}$	One-way hash function	0.00032
$T_{ecm}$	ECC point multiplication	0.0171
$T_{eca}$	ECC point addition	0.0044
$T_{senc}$	Symmetric key encryption	0.0056
$T_{sdec}$	Symmetric key decryption	0.0056
$T_{me}$	Modular exponentiation	0.0192
$T_{fe}$	Fuzzy extractor	0.0171
$T_{R}$	Reproduce operation	0.0171

**Table 5 sensors-20-06860-t005:** Key sizes of the schemes.

Scheme	Key Size
[12]	160 bits
[16]	1024 bits
[17]	1024 bits
[19]	128 bits
[22]	128 bits
**P-I and P-II**	224 bits

**Table 6 sensors-20-06860-t006:** Comparison of the scheme computation cost.

Scheme	[12]	[16]	[17]	[19]	[22]	P-I	P-II
Foreign Network	11*Th* + *TR*	$9Th+TF+2T_{ecm}$	$2T_{ecm}$ + 8*Th*	12*Th* + *TR* + 2*Tme*	2T*ecm* + 3T*h* + 1T*fe*	*4 Th* + *TR*	*2Th* + *TR*
Hub	11*Th*	11*Th*	$4T_{ecm}$ + 18*Th*	10*Th* + *Tsenc*	1T*ecm* + 4T*h*	*8 Th*	*2Th*

**Table 7 sensors-20-06860-t007:** Comparison of the scheme computation time.

Scheme	[12]	[16]	[17]	[19]	[22]	P-I	P-II
Foreign Network	0.02062 s	0.03708 s	0.03676 s	0.05514 s	0.05226 s	0.01838 s	0.01774 s
Hub	0.00352 s	0.00352 s	0.07416 s	0.0088 s	0.01838 s	0.00256 s	0.00064 s

**Table 8 sensors-20-06860-t008:** Comparison of the scheme communication overhead.

Scheme	[12]	[16]	[17]	[19]	[22]	P-I	P-II
Foreign Network	672 bits	1536 bits	1408 bits	896 bits	640 bits	704 bits	480 bits
Hub	512 bits	4096 bits	3968 bits	768 bits	544 bits	480 bits	480 bits

**Table 9 sensors-20-06860-t009:** Storage overhead computation.

Scheme	Stored Data (Foreign Network/SC)	Stored Data (Hub)
[12]	$Q_{i},\;W_{i},\;MID_{i}\;\approx 60\;bytes$	$r_{g}\;\approx 20\;bytes$
[16]	$A_{i},\;B_{i},\;C_{i},\;TID_{i}\approx 64\;bytes$	$TID_{i},\;RN_{G}\approx 24\;bytes$
[17]	$A_{i},\;B_{i},\;n_{0},\;Y,\;P\;\approx 100\;bytes$	$ID_{i},\;r_{i},\;Honey_{List}\approx 20\;bytes$
[19]	$RID_{i},\;f_{i},\;\tau \;\approx 56\;bytes$	$K_{j}\;\approx 20\;bytes$
[22]	$U_{priv}^{\prime },\;t,\;\tau_{i},\;X_{i\;}\approx 56\;bytes$	$K_{priv}\approx 20\;bytes$
P-I	$F_{\left(i+1\right)},\;G\;\approx 40\;bytes$	rand ≈20 bytes

**Table 10 sensors-20-06860-t010:** Comparison of scheme security requirements.

Features	[12]	[16]	[17]	[19]	[22]	Ours
Offline/online shared secret guessing	×	×	×	×	×	✓
Anonymity	×	×	×	×	×	✓
Brute force attack	×	✓	✓	✓	×	✓
*FN*-*SN* Replay attack	✓	✓	✓	✓	✓	✓
*FN*/*HN* Impersonation	×	×	✓	✓	✓	✓
Session hijacking	×	×	×	×	✓	✓
Blockchain data transmission	N/A	N/A	N/A	N/A	N/A	✓
Integrity	✓	✓	✓	✓	✓	✓
Eavesdropping attack	×	×	✓	✓	✓	✓
Re-authentication	N/A	N/A	N/A	N/A	N/A	✓
Un-traceability	✓	×	×	×	✓	✓
Collision attack	-	✓	✓	✓	×	✓
User revocation	N/A	✓	N/A	✓	N/A	✓
Jamming attack	×	×	×	×	✓	✓
Stolen smartcard	×	×	×	×	×	✓

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
