# Peer review of "A Lightweight Three-Factor Authentication Scheme for WHSN Architecture"

_sensors, 2020, doi:10.3390/s20236860_

Round 1

Reviewer 1 Report

The authors propose a three-factor authentication scheme with better system confusion for Wireless Healthcare Sensor Network (WHSN) due to parametric multiplexing characteristics, a hashing function, and a larger key size levitate security and achieve connected node anonymity. His main contributions are cryptanalysis, lightweight three-factor authentication and re-authentication schemes, validation of the improved authentication scheme, key agreement, security, and efficiency calculation.

  1. The related works section is complete, but the authors could delve into other types of interfering attacks, such as eg, jamming for example.
  2. The algorithm on page 13 looks messy and difficult to understand.
  3. The authors take into account performance metrics such as computational cost and overhead. However, they could analyze other parameters related to the energy and capacity of the system.
  4. The application of blockchain in the communication model is not very clear.

Author Response

Referee 1 comments:

Thank you very much for giving me the opportunity to resubmit a modified version of our draft titled “a lightweight three-factor authentication scheme for WHSN architecture”, we appreciate the time and effort that you put on our manuscript to give us your valuable comments.

Comment 1: The related works section is complete, but the authors could delve into other types of interfering attacks, such as eg, jamming for example.

Response 1: Thank you for this comment and we agree on this. Therefore, we applied the following changes on our manuscript:

  • In page 4 lines (144-150): we included some references in the related work about jamming attack.

One of the significant issues that faces the IoT authentication structures is jamming attack, when the intruder  sends jamming signal during the update of authentication values, and parameters (Mbarek, Ge, and Pitner 2020). In this context, two authentication schemes proposed by (Shen et al. 2018; Tewari and Gupta 2017) that employed simple operations such as hash, XOR, and random number generators. Their schemes focused on time duration of the session, mutual random number generation, and keeping the latest identities of the communicating entities to increase the protection against jamming attacks.

  • In page 9 lines (332-334), and page 10 lines (356-357): we refer to our schemes features that make it robust against jamming attack.

  • In page 17, 18 lines (610-617): We made informal analysis about how our scheme is protected against jamming attack in subsection 6.1.12.

Jamming attack

Intruder  tries to disrupt the authentication process by generating a jamming signal to prevent the exchanging of some parameters during the communication. In our scheme, we enabled FN and HN to generate two random numbers that aided the key establishment. The last generated key and identities are used in the creation of the new parameters. So, the Intruder  needs to be aware of the formed session keys, identities, and random values to generate a successful jamming attack. Also, our scheme is protected by a timestamp that prevents the attacker from using old parameters after a long-time passage because the scheme will halt the expired session.

Comment 2: The algorithm on page 13 looks messy and difficult to understand.

Response 2: Thank you for this comment and we agree on this. So, we modified our manuscript accordingly:

  • In pages 12 and page 13: we modified the algorithm and we simplified its contents to make it easier to be understood in figure 3, and figure 4.
  • In Page 9, we added a new figure.

FN (USER)

HN(Gateway)

Enter  and

Input biometric  

Calculate

 Formula (7)               

 Formula (8)

         { 

Delete H from SC

Store in SC.

Produce random value rand

Calculate

 Formula (9)

     Formula (10)

  Formula (11)

Hide rand in a masking formula

 Formula (12)

Store  in SC

}

Comment 3: The authors take into account performance metrics such as computational cost and overhead. However, they could analyze other parameters related to the energy and capacity of the system.

Response 3: Thank you for this interesting comment to enhance our manuscript and we agree on this. so, we updated our manuscript accordingly:

In page 29 lines (867-879): we calculated our scheme capacity and we added a table in subsection 7.

Storage Overhead

We determined the storage cost of our work in contrast to (Mo and Chen 2019; Y. Park and Park 2016; C. Wang, Xu, and Sun 2017; Ali et al. 2020)schemes to analyze the schemes’ capacities. Assuming that each function and parameter of the following have different storage bytes such that hash, ECC, AES symmetric, RSA asymmetric, parameters identifications, random number, and time are 20, 20, 20, 20, 4, 4, 16 bytes, respectively, and the prime  in  is 20 bytes. The suggested scheme requires storage for the stored arguments  that results in  for the smartcard, and  requires 20 bytes for the gateway. The storage cost distinguishes our scheme from others because it is the lowest of all on the smart card side. Moreover, the number of stored security parameters in the proposed structure will provide better security among other schemes, as shown in Table 9 and Table 10.

Scheme

Stored data (Foreign network/SC)

Stored data (Hub)

(Y. Park and Park 2016)

(C. Wang, Xu, and Sun 2017)

(Mo and Chen 2019)

(Ali et al. 2020)

(Yu and Park 2020)

P-I

In page 29 lines (879-889): we added the energy consumption analysis.

From table 6, we compared our proposed scheme computation cost to (Mo and Chen 2019; Y. Park and Park 2016; C. Wang, Xu, and Sun 2017; Ali et al. 2020) schemes and we identified that our scheme performs better than all in both foreign network and hub node sides. (Y. Park and Park 2016)scheme contains 20 hashes, a fuzzy extractor, and 2 ECC point multiplications. (C. Wang, Xu, and Sun 2017) requires 8 hashes, and 26 ECC point multiplications. (Mo and Chen 2019) takes 22 hashes, a reproduction operation, 2 Modular exponentiations, and 1 symmetric encryption. Similarly, (Ali et al. 2020) scheme needs 7 hashes, 3 ECC point multiplications, and a fuzzy extractor. (Yu and Park 2020) takes 22 hashes, and a reproduction function. In comparison to the proposed scheme, our authentication protocol requires 12 hashes along with 1 reproduction function, and the re-authentication protocol requires 4 hashes and 1 reproduction function. This manifests that our scheme has a lower computational cost and low energy consumption.

Comment 4: The application of blockchain in the communication model is not very clear.

Response 4: Thank you for this valuable comment to enhance our manuscript, and we agree on this. Therefore, we modified our manuscript accordingly:

  • In page 14, 15 lines (468-510): we explained in details that (Y. Park and Park 2016) doesn’t include a mechanism to store data securely and how the solutions proposed by (J. Wang et al. 2020) and (Garg et al. 2020) will enhance their scheme in subsection 5.7. as well as we added a new figure to describe the process.

      Secure data transmission via blockchain

In (Yu and Park 2020) scheme, there is no defined strategy to protect the stored data for retrieval or other usages after successful authentication. Since most of the WHSN structures are based on main centralized data storage that is accessible by the assigned doctor. So, this could put patient information in danger due to this source of error. Whereas, the blockchain adds-up the data to blocks and splits them. Therefore, the integrity of the data is kept, each transaction is encrypted. And access control policies guaranteed privacy. Several methods were proposed to aid the purpose of electronic patient record establishing along with patient identity tracking. In the case of the government authorities who want to evaluate the medical facility service or measure the spread of a disease, so the authorities need to have access to all the citizens’ information.

We adopted (J. Wang et al. 2020) method who proposed a Hyperledger blockchain which supports consensus algorithms that only permit the authenticated patients, and communications, and only accept the reserved as well as confidential transactions. The Hyperledger blockchain consists of the transaction log that tackles all the changes made to the connections and changes the value of the world state. The blocks are built by a collection of transactions sent to the evaluator peer to simulate it, vote on it, and approve it. The structure of communication, electronic contracts, access policies are stored in the business network that the user can interact with from a mobile application connected to a server, where all the communication are encrypted by hashing to be able to access the blockchain for data storage and retrieval.

  Another method is discussed by (Garg et al. 2020) that utilized the blockchain technology to store the individual data safely in the cloud.  The sensor nodes contain some data that needs to be stored in the gateway safely for another retrieval or processing. The sensor sends encrypted data with the shared key to the foreign network along with the current timestamp. The foreign network node checks the timeliness and decrypts the data to get the information, then encrypts the data again with its pre-shared key to be sent to the hub node. The hub node decodes the info and checks the timestamp for validity to start building a data block. The block is added to the blockchain when all the communicated entities agreed upon the block contents in peer to peer cloud server network. After successfully gathering a group of valid data, the hub node starts to build transaction values and adds them together in one block to enable the system manager to create a blockchain of data for storage, deletion, update, and retrieve. The proposed method suggested the usage of cryptographic hash to encode the transmitted blocks and compute the ‘‘Merkle tree root’’ (MR) for the tree building. MR is a technique used in cryptocurrency to assure the data integrity in a peer-to-peer network structure. All the block information such as block owner, block payload, etc are computed with the current block hash (CBHash). The hub node embeds the hashed identity of the user and sends the block of data to the system manager which uses ‘‘Ripple Protocol Consensus Algorithm (RPCA)’’ (X. Wang et al. 2019) for node verification and addition. Suppose that a user wants to access some data from a specific block, the user has to log in successfully to the connected hub node. So, as the hub node uses the user key that matches the user identity from the block, performs a hash function on data, decrypts the encrypted data to extract the hashes values and compare them with the computed hash for integrity check. Then, the hub node transmits the data to the user and the user decrypts the data with his/her key to retrieve the information from the block, as depicted in figure 5.

Reviewer 2 Report

Based on the previous scheme previously applied to wireless healthcare sensor network (WHSN), the authors propose a three-factor authentication scheme that includes initialization, authentication, re-authentication, secure node addition, user revocation, and secure data transmission via blockchain technology. The simulation results show that the new scheme has better security and less communication overhead. This paper needs at least major revision, if not rejection

  1. The structure of the paper is loose and the logic is not clear. I can hardly get what the paper is trying to say. There are serious problems with the presentation of the paper and it is very unclear. The paper is too long to read. I suggest that the paper be retried after revision.
  2. Lack of Explicit Problem Formulation. Moreover, it makes it even harder to understand the problem complexity analysis as well as algorithm design. Therefore, I suggest the authors rigorously define their objective functions and all relevant constraints.
  3. Figure 2 and figure 3 directly copy the content of Yu and park strategy, which is complex and difficult to understand. The author should summarize it in his own language.
  4. Section 4.1.1 describes the process of smart card attacks, but the author does not analyze what kind of defects of Yu and park strategy lead to this result.
  5. Figures 4 and 5 are misplaced and their contents are confusing.
  6. In Section 5.7, the description of the application of blockchain is not detailed enough.
  7. Some problems have been addressed by the authors, however, the novelty of this paper seems still limited. The reviewer strongly suggests that the theoretical analysis of the system performance should be added to improve the quality of this paper.

Author Response

Referee 2 comments:

Thank you very much for giving me the opportunity to submit a revised version of our manuscript titled “a lightweight three-factor authentication scheme for WHSN architecture”, we’re grateful to the reviewer valuable comments, the time, and effort that have been made to improve the quality of our article.

Comment 1: The structure of the paper is loose, and the logic is not clear. I can hardly get what the paper is trying to say. There are serious problems with the presentation of the paper and it is very unclear. The paper is too long to read. I suggest that the paper be retried after revision.

Response 1: Thank you for the insightful comment to enhance our manuscript and we agree on this. So, we updated our manuscript as follows: we added few more tables and figures to help us organize our paper to be more presentable and give it a vivid structure. We reduced the number of the pages by omitting some graphs and add some tables to arrange the paper contents in a clearer way.

  • In page 9: we added new figure regarding our registration phase.

FN (USER)

HN(Gateway)

Enter  and

Input biometric  

Calculate

 Formula (7)               

 Formula (8)

         { 

Delete H from SC

Store in SC.

Produce random value rand

Calculate

 Formula (9)

     Formula (10)

  Formula (11)

Hide rand in a masking formula

 Formula (12)

Store  in SC

}

  • In page 18: we added new table regarding BAN logic notations

Symbols

Description

M|≡ N

M trusts N

MN

M sees N

M| ∼ N

M once responded N

M|=⟩ N

M governs N

#(N)

N is new

NB

N is merged with B

NB

N is encrypted by B

N Q

K is shared secret between N and Q

Pre-shared key used in connection

  • In page 29: we added new table about storage computation in our scheme.

Scheme

Stored data (Foreign network/SC)

Stored data (Hub)

(Y. Park and Park 2016)

(C. Wang, Xu, and Sun 2017)

(Mo and Chen 2019)

(Ali et al. 2020)

(Yu and Park 2020)

P-I

Comment 2: Lack of Explicit Problem Formulation. Moreover, it makes it even harder to understand the problem complexity analysis as well as algorithm design. Therefore, I suggest the authors rigorously define their objective functions and all relevant constraints.

Response 2 Thank you for this comment and we agree on this. Therefore, we modified our manuscript as follows:

  • In pages 2- 3 line (57-88) we added our contributions, objectives, and motivations.

Comment 3: Figure 2 and figure 3 directly copy the content of Yu and park strategy, which is complex and difficult to understand. The author should summarize it in his own language.

Response 3: Thank you for this comment and we agree on this. Therefore, we modified our manuscript as follows:

  • In pages 5- 6 line (196-223): Figure 2 and figure 3 have been remove. We state that we summarized Yu and Park scheme in our own words, and we clarified its weaknesses in conjunction with our cryptanalysis. In subsection 4.2 and subsection 4.2.

Registration phase of Yu and Park scheme

In the registration phase of (Yu and Park 2020), the user and GWN communicate with one another to produce username, password, biometric, and smartcard values:

  • The user , inputs his/her password, username, and biometric, extract the biometric features using reproduction function and send those value over a secure channel to GWN.
  • GWN produces random value , calculates identification values , , , and to store {Qi, Wi, MIDi} in the SC, and save in secure database. The number of saved parameters in the smartcard causes a weakness, that the attacker can seize to exploit the system parameters, by performing a smartcard impersonation attack along with database hijacking to retrieve the random value and user biometric.  

 Authentication phase of Yu and Park scheme

In the authentication phase of (Yu and Park 2020), the user and GWN authenticate each other along with the sensor to agree on the next session key as follows:

  • Client inputs his/her username, password, and biometric in the smartphone, and checks the user identity before generating a current random value  along with a timestamp.
  • The mobile masks the following parameters such as original user identity , the hidden user identity , user masked identity in the smartcard , and the masked .
  • The GWN node receives the parameters, checks the timestamp to avoid replay attack, and retrieves the random values from the masked without checking if the random value had been used before. This step might rise a vulnerability in the scheme in case of a user node impersonation attack. 
  • The GWN shares the hidden values with the sensor for further authentication to the user node, and to support the next session key generation.
  • authenticates both GWN and , and generates new random nonce to produce the next pre-shared key.
  • Both GWN and receive the new parameters to recover the generated values and save the new session key.   

Comment 4: Section 4.1.1 describes the process of smart card attacks, but the author does not analyze what kind of defects of Yu and park strategy leads to this result.

Response 4: We appreciate this comment and we accept this. Thus, we review our paper by discussing the weaknesses of Yu and Park scheme that we based our cryptanalysis on.

  • In pages 6 line (202-205): we pointed the weakness point in the registration phase which is the stored parameters in the smartcard, and we reduced them in our scheme.

  • In pages 6 line (214-217): we pointed the weakness point in the authentication phase which is the absence of random value checking in the hub node side and we add this feature in our scheme.

  • In pages 6 line (225-229): we mentioned the vulnerabilities in Yu and Park scheme to validate our cryptanalysis in subsection 4.3.

Cryptanalysis of Yu and Park scheme

From the above, We deliberated the flaws for (Yu and Park 2020) structure in both registration and authentication phases which are: the number of authentication parameters in the smartcard along with the absence of random number checking in the GWN. Those two weaknesses allow the intruder to impersonate the user in the lost smartcard attack and weaken the anonymity.

  • In pages 16 line (553-561): we mentioned the how our scheme can prevent smart card attack by adding locking and encrypting policy to the smart card.

Stolen smartcard attack

In (Yu and Park 2020) scheme, they didn’t specify a method to prevent brute force attacks in case of the lost smart card. Since their paper didn’t mention the concept of encrypting and locking smartcard data information with user biometric or password. Therefore, we suggest in the cryptanalysis to reduce the number of parameters, random number checking, as well as smart card blocking policy after three times error in entering the authentication biometric, and password. Moreover, encrypting and locking the card information with a password, and user biometric at each time to authenticate the user to the smartcard will guarantee the tamper-resistant feature when the card is lost. Thus, our scheme prevents stolen smartcard attack.

Comment 5: Figures 4 and 5 are misplaced and their contents are confusing.

Response 5: Thank you for this comment and we accept this comment. Therefore, we modified our manuscript accordingly:

In pages 12 and page 13: we modified the algorithm and we simplified its content to make it easier to understand in figure 3, and figure 4.

Comment 6: In Section 5.7, the description of the application of blockchain is not detailed enough.

Response 6: We appreciate your insightful comment, we accept it and we apply the changes in our manuscript as follows:

In page 14 lines (469-491): we explained in details that (Y. Park and Park 2016) doesn’t include a mechanism to store data securely and how the solutions proposed by (J. Wang et al. 2020)  and (Garg et al. 2020) will enhance their scheme in subsection 5.7. as well as we added a new figure to describe the process.

    Secure data transmission via blockchain

In (Yu and Park 2020) scheme, there is no defined strategy to protect the stored data for retrieval or other usages after successful authentication. Since most of the WHSN structures are based on main centralized data storage that is accessible by the assigned doctor. So, this could put patient information in danger due to this source of error. Whereas, the blockchain adds-up the data to blocks and splits them. Therefore, the integrity of the data is kept, each transaction is encrypted. And access control policies guaranteed privacy. Several methods were proposed to aid the purpose of electronic patient record establishing along with patient identity tracking. In case of the government authorities wants to evaluate the medical facility service, or measure the spread of a disease, so the authorities need to have access to all the citizens information.

We adopted (J. Wang et al. 2020) method who proposed a Hyperledger blockchain which supports consensus algorithms that only permit the authenticated patients, and communications, and only accept the reserved as well as confidential transactions. The Hyperledger blockchain consists of the transaction log that tackles all the changes made to the connections and changes the value of the world state. The blocks are built by a collection of transactions sent to the evaluator peer to simulate it, vote on it, and approve it. The structure of communication, electronic contracts, access policies are stored in the business network that the user can interact with from a mobile application connected to a server, where all the communication are encrypted by hashing to be able to access the blockchain for data storage and retrieval.

  Another method is discussed by (Garg et al. 2020) that utilized the blockchain technology to store the individual data safely in the cloud.  The sensor nodes contain some data that needs to be stored in the gateway safely for another retrieval or processing. The sensor sends encrypted data with the shared key to the foreign network along with the current timestamp. The foreign network node checks the timeliness and decrypts the data to get the information, then encrypts the data again with its pre-shared key to be sent to the hub node. The hub node decodes the info and checks the timestamp for validity to start building a data block. The block is added to the blockchain when all the communicated entities agreed upon the block contents in peer to peer cloud server network. After successfully gathering a group of valid data, the hub node starts to build transaction values and adds them together in one block to enable the system manager to create a blockchain of data for storage, deletion, update, and retrieve. The proposed method suggested the usage of cryptographic hash to encode the transmitted blocks and compute the ‘‘Merkle tree root’’ (MR) for the tree building. MR is a technique used in cryptocurrency to assure the data integrity in a peer-to-peer network structure. All the block information such as block owner, block payload, etc are computed with the current block hash (CBHash). The hub node embeds the hashed identity of the user and sends the block of data to the system manager which uses ‘‘Ripple Protocol Consensus Algorithm (RPCA)’’ (X. Wang et al. 2019) for node verification and addition. Suppose that a user wants to access some data from a specific block, the user has to log in successfully to the connected hub node. So, as the hub node uses the user key that matches the user identity from the block, performs a hash function on data, decrypts the encrypted data to extract the hashes values and compare them with the computed hash for integrity check. Then, the hub node transmits the data to the user and the user decrypts the data with his/her key to retrieve the information from the block, as depicted in figure 5.

Comment 7: Some problems have been addressed by the authors; however, the novelty of this paper seems still limited. The reviewer strongly suggests that the theoretical analysis of the system performance should be added to improve the quality of this paper.

Response 7: Thank you for pointing out this issue and we accept the comment. However, our contribution can be highlighted in the performance analysis section where we showed that our technique outperforms other works in computation, communication, and security requirement. Also, we showed that the proposed solution performs better in the foreign network side in term of capacity and consume less energy.

In page 29, 30 lines (854-881): we calculated our scheme computation cost and communication cost and security requirement. We added a table 6,7,8,10.

Scheme

(Y. Park and Park 2016)

(C. Wang, Xu, and Sun 2017)

(Mo and Chen 2019)

(Ali et al. 2020)

(Yu and Park 2020)

P-I

P-II

Foreign network

9Th +TF +2

2+8Th

12Th +TR +2Tme

2Tecm +3Th +1Tfe

11Th + TR

4 Th +TR

2Th +TR

Hub

11Th

4+18Th

10Th +Tsenc

1Tecm +4Th

11Th

8 Th

2 Th

Scheme

(Y. Park and Park 2016)

(C. Wang, Xu, and Sun 2017)

(Mo and Chen 2019)

(Ali et al. 2020)

(Yu and Park 2020)

   P-I

P-II

Foreign network

0.03708 s

0.03676 s

0.05514 s

0.05226 s

0.02062 s

0.01838 s

0.01774 s

Hub

0.00352 s

0.07416 s

0.0088 s

0.01838 s

0.00352 s

0.00256 s

0.00064 s

Scheme

(Y. Park and Park 2016)

(C. Wang, Xu, and Sun 2017)

(Mo and Chen 2019)

(Ali et al. 2020)

(Yu and Park 2020)

P-I

P-II

Foreign network

1536 bits

1408 bits

896 bits

640 bits

672 bits

704 bits

480 bits

Hub

4096 bits

3968 bits

768 bits

544 bits

512 bits

480 bits

480 bits

Features

(Y. Park and Park 2016)

(C. Wang, Xu, and Sun 2017)

(Mo and Chen 2019)

(Ali et al. 2020)

(Yu and Park 2020)

Ours

Offline/online shared secret guessing

×

×

×

×

×

Anonymity

×

×

×

×

×

Brute force attack

×

×

FN-SN Replay attack

FN/HN Impersonation

×

×

Session hijacking

×

×

×

×

Blockchain data transmission

N/A

N/A

N/A

N/A

N/A

Integrity

Eavesdropping attack

×

×

Re-authentication

N/A

N/A

N/A

N/A

N/A

Un-traceability

×

×

×

Collision attack

×

-

User revocation

N/A

N/A

N/A

Jamming attack

×

×

×

×

In page 29 lines (867-879): we calculated our scheme capacity and we added a table in subsection 7.

Storage Overhead

We determined the storage cost of our work in contrast to (Mo and Chen 2019; Y. Park and Park 2016; C. Wang, Xu, and Sun 2017; Ali et al. 2020)schemes to analyze the schemes’ capacities. Assuming that each function and parameter of the following have different storage bytes such that hash, ECC, AES symmetric, RSA asymmetric, parameters identifications, random number, and time are 20, 20, 20, 20, 4, 4, 16 bytes, respectively, and the prime  in  is 20 bytes. The suggested scheme requires storage for the stored arguments  that results in  for the smartcard, and  requires 20 bytes for the gateway. The storage cost distinguishes our scheme from others because it is the lowest of all on the smart card side. Moreover, the number of stored security parameters in the proposed structure will provide better security among other schemes, as shown in Table 9 and Table 10.

Scheme

Stored data (Foreign network/SC)

Stored data (Hub)

(Y. Park and Park 2016)

(C. Wang, Xu, and Sun 2017)

(Mo and Chen 2019)

(Ali et al. 2020)

(Yu and Park 2020)

P-I

In page 29 lines (879-889): we added the energy consumption analysis.

From table 6, we compared our proposed scheme computation cost to (Mo and Chen 2019; Y. Park and Park 2016; C. Wang, Xu, and Sun 2017; Ali et al. 2020) schemes and we identified that our scheme performs better than all in both foreign network and hub node sides. (Y. Park and Park 2016)scheme contains 20 hashes, a fuzzy extractor, and 2 ECC point multiplications. (C. Wang, Xu, and Sun 2017) requires 8 hashes, and 26 ECC point multiplications. (Mo and Chen 2019) takes 22 hashes, a reproduction operation, 2 Modular exponentiations, and 1 symmetric encryption. Similarly, (Ali et al. 2020) scheme needs 7 hashes, 3 ECC point multiplications, and a fuzzy extractor. (Yu and Park 2020) takes 22 hashes, and a reproduction function. In comparison to the proposed scheme, our authentication protocol requires 12 hashes along with 1 reproduction function, and the re-authentication protocol requires 4 hashes and 1 reproduction function. This manifests that our scheme has a lower computational cost and low energy consumption.

Round 2

Reviewer 2 Report

The authors have addressed my concern and it can be accepted now.